# The evolution of manipulative cheating

**Ming Liu[1]\*, Stuart Andrew West[1], Geoff Wild[2]**

[1]Department of Biology, University of Oxford, Oxford, United Kingdom; [2]Department of Mathematics, The University of Western Ontario, London, Canada

**Abstract** A social cheat is typically assumed to be an individual that does not perform a cooperative behaviour, or performs less of it, but can still exploit the cooperative behaviour of others. However, empirical data suggests that cheating can be more subtle, involving evolutionary arms races over the ability to both exploit and resist exploitation. These complications have not been captured by evolutionary theory, which lags behind empirical studies in this area. We bridge this gap with a mixture of game-theoretical models and individual-based simulations, examining what conditions favour more elaborate patterns of cheating. We found that as well as adjusting their own behaviour, individuals can be selected to manipulate the behaviour of others, which we term 'manipulative cheating'. Further, we found that manipulative cheating can lead to dynamic oscillations (arms races), between selfishness, manipulation, and suppression of manipulation. Our results can help explain both variation in the level of cheating, and genetic variation in the extent to which individuals can be exploited by cheats.

## Editor's evaluation

This theoretical paper examines the conditions under which manipulative cheating can evolve. Manipulative cheating is a form of coercion where individuals can benefit by actively manipulating others to help them at a cost to their own fitness. The paper seamlessly integrates a multilevel selective framework and inclusive fitness, with a rigorous analysis of the joint dynamics of selfishness, manipulation, and suppression of manipulation. The results are novel and important, as they help us to better understand the evolution of cooperation and the spectrum of social cheating, potentially opening up new directions for theoretical and empirical work.

**\*For correspondence:**
ming.liu@biology.ox.ac.uk

**Competing interest:** The authors declare that no competing interests exist.

## Introduction

Many social interactions involve a trade-off between investing in behaviours or traits that increase the productivity of the group, versus those that increase an individual's share of group productivity (*Frank, 1998*; *Bourke, 2011*; *Queller and Strassmann, 2018*; Figure 1a). A female social insect larva could become a queen or a worker (*Hamilton, 1972*). A slime mould cell could become a stalk or a spore within a fruiting body (*Strassmann et al., 2000*). A bacterial cell could invest costly resources into producing some 'public good' that benefits the local group of cells, such as iron scavenging siderophore molecules (*Griffin et al., 2004*).

A standard assumption is that individuals can behave more selfishly by investing less in cooperative behaviours that benefit the group, and by taking a larger share of the available resources (*Ghoul et al., 2014*). For example, a slime mould lineage could 'cheat' by having a larger proportion of its cells develop into spore cells, or a bacterial cell could cheat by producing fewer siderophores (*Strassmann et al., 2000*; *Jiricny et al., 2010*). When cheats can invade, theory predicts that this will lead to either uniform cooperation at a lower level or the coexistence of cheats and cooperators (*Queller, 1984*; *Frank, 1994*; *Frank, 1998*; *West and Buckling, 2003*; *MacLean and Gudelj, 2006*; *Ross-Gillespie et al., 2007*; *Gore et al., 2009*; *Frank, 2010c*; *Frank, 2010a*; *Frank, 2010b*; *Patel et al.,*

*2019*). Coexistence is only expected when some factor leads to the fitness of cheats being higher when they are less common (frequency-dependent selection).

In contrast, the empirical data suggests that there can be more complex forms of cheating and resistance to cheating within social groups. For example, in the slime *Dictyostelium discoideum*: (i) lineages appear to be able to evolve resistance to cheating, where they prevent other lineages from obtaining a higher proportion of spore cells (*Buttery et al., 2009*; *Khare et al., 2009*; *Levin et al., 2015*); (ii) genomic data suggests an ongoing evolutionary arms race, and not just selection towards an optimal level of cooperation (*Ostrowski et al., 2015*; *Noh et al., 2018*; *Ostrowski, 2019*). In some ant species, genetic variation appears to be maintained in the extent to which different patrilines contribute to the production of workers in a colony, a phenomenon termed 'royal cheating' (*Hughes et al., 2003*; *Linksvayer, 2006*; *Anderson et al., 2008*; *Hughes and Boomsma, 2008*; *Mitchell et al., 2012*; *Stürup et al., 2014*). In *Pseudomonas* bacteria, strains appear to vary in both the extent to which they can cheat by exploiting the siderophores produced by other strains and the extent to which they can be cheated by other strains (*Kümmerli et al., 2009*; *Bruce et al., 2017*; *Butaité et al., 2017*). In all these cases, cheating appears to involve doing more than just cooperating at a lower level. Other examples of more complex forms of cheating can be found in bacteria, viruses, and replicator RNA (*Pollak et al., 2016*; *Furubayashi et al., 2020*; *Bruce et al., 2021*; *Mizuuchi et al., 2022*).

Our aim is to explain how these more complex forms of cheating can be maintained in a population. As mentioned above, most evolutionary models assume that an individual can control their own investment into selfish and cooperative behaviours – put simply, an individual cheats by investing less in cooperation (*Queller, 1984*; *MacLean and Gudelj, 2006*; *Ross-Gillespie et al., 2007*; *Gore et al., 2009*; *Frank, 2010c*; *Patel et al., 2019*). We investigate another possibility, that individuals can manipulate the behaviour of others. For example, an ant patriline may increase the likelihood that its own larvae are raised as queens, while reducing the likelihood that the larvae of other patrilines are reared. Or a bacteria cell lineage could increase the public goods produced by another lineage. Such 'manipulative cheating' could involve a variety of mechanisms, ranging from coercive communication to more direct influencing.

We develop a series of models to examine whether manipulative cheating can be favoured, and if it will lead to variation in the extent to which individuals can exploit others. In addition, we explore how once manipulation has evolved, there could be selection to evolve resistance or suppression of that cheating. We use a mixture of game theory and simulation, to examine scenarios where resistance can evolve relatively simply, or where there are multiple mechanisms of manipulative cheating that each require a different form of resistance. Our model assumes an asexual haploid population, but we also examine the consequences of recombination. We find that natural selection can maintain a variety of strategies at equilibrium, as well as the possibility for ongoing oscillating dynamics, such as evolutionary arms races. While our focus is on 'manipulation', our more general aim is to investigate scenarios where cheating is more than just 'cooperate less'.

## Materials and methods
### Model overview
Our aim is to investigate the general feasibility of manipulation, in a way that could be applicable across a range of species and does not rely on the specifics of any single species. Consequently, we have chosen a deliberately simple approach to maintain tractability. We use a combination of modelling approaches to examine the evolutionary stability and dynamics of manipulative cheating.

First, we use an equilibrium game theoretical approach to examine under what conditions manipulative cheating could be favoured. Our model is based on the classic tragedy of the commons model (*Frank, 1994*; *Frank, 1996*; *Frank, 1998*; *Dionisio and Gordo, 2006*; *Frank, 2010a*; *Frank, 2010b*), where individuals can invest in their own selfish reproduction, or the productivity of the group (*Figure 1a*; Appendix 1—1.2). We extend this model by assuming that: (i) individuals can invest effort into manipulating the behaviour of others to make them cooperate at a higher rate (behave less selfishly; *Figure 1b*); and (ii) individuals can also invest effort into suppressing (blocking) the extent to which they are manipulated by others (*Figure 1c*). We then test the robustness of our game-theoretical results by comparing them with individual-based simulations.

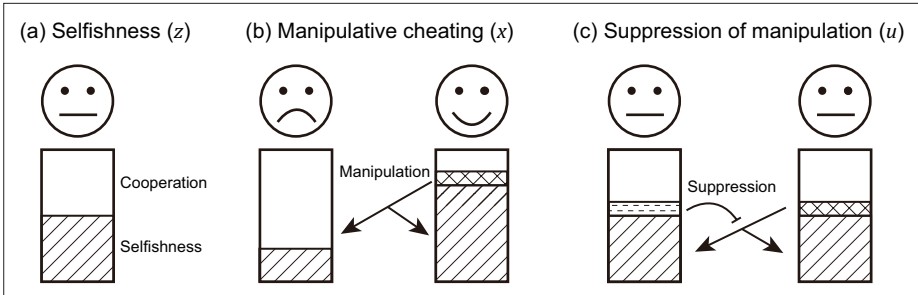

**Figure 1.** Schematic view of the three evolving traits in our model. Each individual can put effort into three actions: (**a**) invest resources into reproduction ($z_i$; represented by stripe texture) or group productivity (average cooperation; $1 - \overline{z}$); (**b**) manipulate others ($x_i$; represented by the grid texture), to make them invest more into group productivity; and (**c**) suppress (block) the manipulation from others ($u_i$; represented by horizontal dashed texture).

Second, we examine the consequence of manipulative cheating when multiple different mechanisms of manipulation can evolve, which each requires a different mechanism to suppress. We consider cases where traits are introduced at the same time point or different time points (mutation) and focus on the temporal trait dynamics. As multiple mechanisms could generate complications between traits which cannot be easily presented by equilibrium analysis, we focus on dynamic results from individual-based simulations.

Our general aim is to investigate scenarios where cheating can be more than just 'cooperate less'. We do this by developing models focused on the special case of manipulation and its suppression. Other possibilities would include developing coevolutionary genetic models to look at genotype-specific cheating-resistance, analogous to gene-for-gene or matching alleles models of host-parasite coevolution, or to model a different form of coercion (*Sasaki, 2000*; *Agrawal and Lively, 2002*).

## Model design

We assume a haploid and asexual population inhabiting an infinite number of identical habitat patches, where each patch supports a single social group and lets social interactions take place. We also assume the fitness of each individual is proportional to the productivity of the social group multiplied by the individual's share of group productivity, and all offspring compete globally for patches in the next generation. Within each social group, each individual divides its effort into cooperation with group members, which aids group productivity or selfishly reproducing (gaining a greater share of the group productivity). We consider three coevolving traits:

*Selfishness* ($z$): We assume that when individuals put more effort into reproduction ($z$), they: (i) obtain a larger share of the group productivity; (ii) put less effort into cooperation, which reduces group productivity. Specifically, if a focal individual $i$ invests $z_i$, into reproduction, and the group average of all individuals in the group is $\overline{z}$ ($0 < z < 1$). In other words, the parameter $z$ measures the level of 'selfishness' (*Figure 1a*; *Frank, 1994*; *Frank, 1996*; *Frank, 1998*; *Dionisio and Gordo, 2006*; *Frank, 2010a*; *Frank, 2010b*).

*Manipulative cheating* ($x$): We assume that a focal individual can invest $x$ in manipulation, which causes other individuals to invest fewer resources into their own reproduction, and instead invest those resources into the reproduction of the focal individual (manipulator; $0 < x < 1$; *Figure 1b*). This manipulation requires effort and so reduces the resources that the manipulator contributes to group productivity (by $\overline{x}$). Manipulation $x$ therefore takes reproduction from others, whereas selfishness $z$ shifts an individual's allocation between cooperation and reproduction. Consequently, manipulation is redistributing the share of all individual's reproduction while z is changing the effort spent in reproduction. For simplicity, we have assumed there is no 'loss' in reproductive effort during the process of manipulation, but we found similar results when manipulation also leads to a personal reproduction cost (Appendix 1—3).

*Suppression of manipulative cheating* ($u$): We add the possibility for individuals to suppress (block) manipulation ($0 < u < 1$; *Figure 1c*). When suppression exists, a manipulator can only take a proportion

$1 - u^b$ of reproduction, where $b$ is a shape parameter (see below). Like the other two traits, suppression takes effort and reduces the effort put into group productivity (by $\bar{u}$.

Combining the effects of all three traits lead to:

$$w_i = \frac{z_i + \left(1 - u^b_{-i}\right)x^a_i z_{-i} - \left(1 - u^b_i\right)x^a_{-i}z_i}{\bar{z}}\left(1 - \bar{z} - \bar{x} - \bar{u}\right), \tag{1}$$

where $z_i$ is the selfishness of the focal individual, $\left(1 - u^b_{-i}\right)x^a_i z_{-i}$ is the gain from manipulating others, $\left(1 - u^b_i\right)x^a_{-i}z_i$ is the loss from being manipulated, and $1 - \bar{z} - \bar{x} - \bar{u}$ is the group productivity. Because the sum of gain and loss across all individuals in the group is zero, the average effort in reproduction is just $\bar{z}$. Thus, the first term is the relative share of group productivity of the focal individual (i.e. $\frac{z_i + \left(1 - u^b_{-i}\right)x^a_i z_{-i} - \left(1 - u^b_i\right)x^a_{-i}z_i}{\bar{z}}$). In other words, our manipulative cheating trait involves an individual actor causing a recipient (other group members) to behave less selfishly (more cooperatively), in a way that is beneficial to the actor, but costly to the recipient. This contrasts with previous theory which examined the consequences of changing the benefit and cost of social behaviour (*Frank, 1995*), or when relatives are manipulated to perform a social behaviour that provides an indirect benefit to the actor (*González-Forero and Gavrilets, 2013*; Appendix 1—1.3). We provide a more detailed derivation of *equation 1* in Appendix 1—1.1. In the special case of no manipulation or suppression, *equation 1* simplifies to the classic tragedy of the commons model (*Frank, 1994*; *Frank, 1996*; *Frank, 1998*; *West et al., 2002*; *Dionisio and Gordo, 2006*; *Frank, 2010a*; *Frank, 2010b*).

We allow the effects of manipulation and suppression to be nonlinear by the shape parameters, $a$ and $b$. Nonlinearities can arise for a number of reasons, such as mechanisms of molecular diffusion, or interference between the behaviour of different individuals (*Müller et al., 2006*; *Wiley, 2013*; *Rogers et al., 2017*). The parameter $a$ determines whether the benefit from increased investment into manipulation is decelerating ($a < 1$), linear ($a = 1$) or accelerating ($a > 1$). Similarly, the shape parameter $b$ determines the shape of the benefit of increased investment into suppression. We have focused on the decelerating forms because this will often be the case for biological traits (Appendix 1—5.1).

We use two approaches to analyse the model with different assumptions about recombination, or the degree of linkage, between traits. Our first analysis uses the neighbour modulated fitness of *Taylor and Frank, 1996*, where one trait is modified at a time to find the convergent stable strategy (Model Analysis and Appendix 1—1.1–1.2). The analysis assumes all three traits evolve independently; this is equivalent to a model that assumes a sexual diploid with additive interactions between alleles at a given locus and without recombination (*Taylor, 1996*; *Day and Taylor, 1998*; *Kisdi and Geritz, 1999*; *Geritz and Kisdi, 2000*). The second approach is through individual-based simulations, where recombination process can be explicitly modelled through how traits are inherited between generations (Appendix 1—7). The main simulation results also assume the absence of recombination. Nevertheless, we relax the assumption and include recombination by allowing haploid asexual individuals the chances to swap traits in Appendix 1—4.4 and 4.8.

We found it useful to think about manipulation, suppression, and *equation 1* with a lottery-ticket metaphor. Imagine reproduction as the buying of a lottery ticket – producing an offspring, that has some chance of 'winning' (reproducing) in the next generation. Individuals can either invest resources into buying the lottery tickets produced by the group ($z$; selfish), or invest resources into helping the group produce more lottery tickets ($1 - z$; cooperation). Each group produces its own lottery tickets and more tickets means a higher chance for the focal group to win. Being more selfish (higher $z$) means obtaining a larger fraction of the lottery tickets produced by the group, but also putting fewer resources into producing tickets (and so the group procures fewer tickets). In contrast, manipulation (higher $x$) allows individuals to steal lottery tickets from others within their group. Suppression, $u$, would involve blocking lottery tickets from being stolen. We consider biological examples of manipulation and suppression in the discussion section.

## Results

### Model analysis

We seek the evolutionarily stable strategy (ESS), which is the individual strategy that cannot be invaded by any other rare strategy. Using the neighbour-modulated fitness approach of *Taylor and Frank, 1996*, we find:

$$
\begin{cases}
\Delta W_z = \left( \bar{R}_{-i} \dfrac{-\left( \frac{n-1}{n} - x^a \left(1-u^b\right) \right)}{z} + \dfrac{\frac{n-1}{n} - x^a \left(1-u^b\right)}{z} \right) \left(1 - z - x - u\right) - \bar{R} \\[2ex]
\Delta W_x = \left( \bar{R}_{-i} \left( -ax^{a-1} \right) \left(1-u^b\right) + ax^{a-1} \left(1-u^b\right) \right) \left(1 - z - x - u\right) - \bar{R}' \\[2ex]
\Delta W_u = \left( \bar{R}_{-i} \left( -x^a \right) bu^{b-1} + x^a bu^{b-1} \right) \left(1 - z - x - u\right) - \bar{R}
\end{cases}
\tag{2}
$$

where traits are analysed one at a time, where $\Delta W_z$ means the change in fitness due to a slight increase in selfishness in the focal individual, and analogous for $\Delta W_x$ and $\Delta W_u$. In addition, $\bar{R}_{-i}$ denotes the relatedness of focal individual to all other group members, $\bar{R}$ indicate the whole-group relatedness (*Pepper, 2000*; Appendix 1—1.2). The two relatedness terms can be linked— for example, if the group size is $n$, then $\bar{R} = 1/n + \left( (n-1)/n \right) \bar{R}_{-i}$. For simplicity, we assume all traits only operate within the group so that $\bar{R}_{-i}$ does not differ between traits. More general form allowing for cases when different traits are at different social scales is considered in the derivation in Appendix 1—1. We also assume each trait evolves independently and there is no allele-level interaction between each trait. Our analyses identify the candidate ESS or convergently stable strategy (*Eshel, 1983*; *Christiansen, 1991*), which can be used to make comparative statics predictions (*Frank, 1998*).

Each of the three equations in *Equation 2* represents the change to the inclusive fitness of an individual that occurs when the individual increases the level at which it expresses the focal trait (*Hamilton, 1964a*; *Hamilton, 1964b*). Because all three expressions share similar structure, we can simplify them into

$$
\Delta W_{trait} = \left( \bar{R}_{-i} \left( -A \right) + A \right) \left(1 - z - x - u\right) - \bar{R},
\tag{3}
$$

where $A$ is different for each trait and subscript 'trait' means the type of trait we are analysing, which could be $z$, $x$, or $u$. The first term on the right, $-A$, represents the relatedness-weighted reduction in neighbours' share of group productivity due to an individual's selfishness or manipulation (an inclusive-fitness cost; e.g. $\dfrac{-\left( \frac{n-1}{n} - x^a \left(1-u^b\right) \right)}{z}$ for selfishness). The second term, $A$, represents the increase in personal share of group productivity made by a selfish or manipulative individual (an inclusive-fitness benefit; e.g. $\dfrac{\frac{n-1}{n} - x^a \left(1-u^b\right)}{z}$ for selfishness). The third term, group productivity, represents a relatedness-weighted reduction in fitness of the average group member, self-included, owing to selfishness, manipulation, or suppression (i.e. $1 - z - x - u$). The cost in group productivity is spread equally among all individuals in the group, including the actor. The last term, $\bar{R}$, represents the indirect cost of reducing the share of group productivity obtained by relatives.

*Equation 2* could not be solved analytically, and so we found solutions with an iterative numerical method (*Figure 2*; Appendix 1—6.2). We tested the robustness of our solutions with an individual-based simulation, which includes mutation and drift. Our simulations and numerical predictions showed close agreement (*Figure 2*; Appendix 1—7).

We found that the ESS selfishness, manipulation, and suppression are all predicted to show negative relationships with relatedness (*Figure 2*). At a lower relatedness, individuals are predicted to: (i) selfishly keep a higher fraction of their resources for themselves (as predicted by the classic tragedy of the commons model); (ii) invest more heavily in manipulating others to help them (as also predicted in a model without suppression; Appendix 1—2); (iii) invest more heavily in suppressing manipulation perpetrated by others (when manipulation is at higher levels). These predictions qualitatively hold for a broad range of parameter space, with linear or non-linear returns on increased investment in manipulation and suppression, and show that manipulative cheating as well as its suppression can be favoured by natural selection.

We predicted that the level of selfishness was greater than the level of manipulation, which was greater than the level of suppression ($z > x > u$). To a large extent this reflects that benefit of

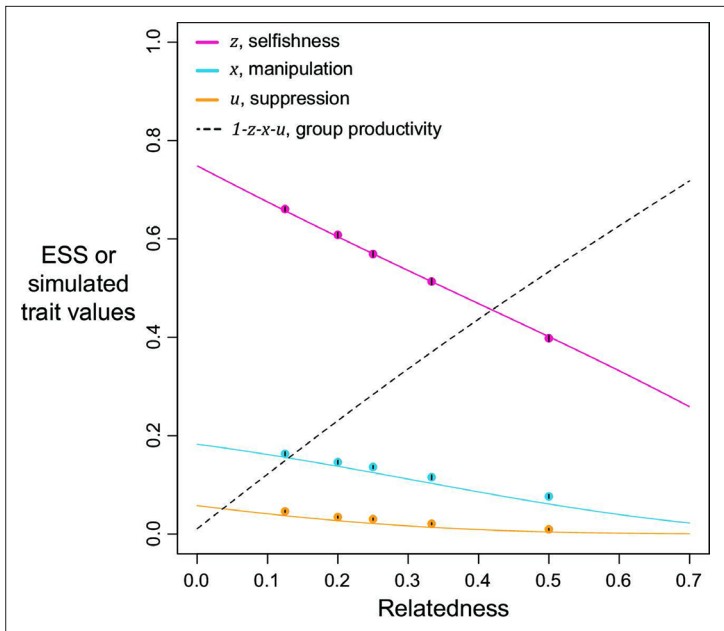

**Figure 2.** Selfishness, manipulation, and suppression of manipulation. The panels show the ESS predictions and simulation results for our three-trait model. The solid lines show the ESS predictions, and the dots show the results of our individual-based simulations ($a = b$=0.5). The black segments within each dot show the standard deviation of trait values of the last 10% duration of 10 repeated simulations (Appendix 1—7).

manipulation depends upon the level of selfishness, and the benefit of suppression depends upon the level of manipulation. However, the quantitative predictions depend on details such as the specific values of the shape parameters (see below), and these relations need not always hold. Our simulations, which involved mutations, led to relatively static predictions when there is only one mechanism to manipulate and suppress, in agreement with our analytical results, supporting the results as ESSs (dots versus lines in *Figure 2*).

The quantitative ESS predictions depend upon the shape parameters (*Figure 3*). When the shape parameter of manipulation, $a$, is more decelerating, selfishness becomes lower, suppression becomes higher, and manipulation becomes higher until suppression evolved (*Figure 3a–c*). Secondly, when the shape parameter of suppression, $b$, is more decelerating, selfishness becomes higher, manipulation becomes lower, and suppression becomes higher until there is no manipulation to suppress (*Figure 3g–i*). Thirdly, when the two shape parameters covary together, the impact on ESS trait values is a combination of the two forementioned effects (*Figure 3d–f*). Both manipulation and suppression can be favoured across a wide range of parameter values.

## Multiple mechanisms for manipulative cheating and suppressing

In our above model, we have focused on evolutionary equilibrium, assuming there is only a single mechanism to manipulate others. In nature, manipulation could take many forms, and each requires a different form of suppression. This could lead to complicated dynamics, such as a genetic arms race, where different forms of manipulation arise and are then suppressed over time. Because the rise and fall of trait values are potentially transient, we move away from game-theoretical analysis and focus on non-equilibrium scenarios through individual-based simulations. We present two different cases here: one with multiple mechanisms where all are introduced at the same time, and the other with multiple mechanisms where new mechanisms are introduced to the population at different time points (via mutation).

Assuming there are $k$ mechanisms to manipulate and suppress, fitness is given by

$$\begin{aligned} w_i = &\left( z_i + \left( 1 - u_{1,-i}^b \right) z_{-i} x_{1,i}^a - \left( 1 - u_{1,i}^b \right) z_i x_{1,-i}^a + \left( 1 - u_{2,-i}^b \right) z_{-i} x_{2,i}^a \\ &- \left( 1 - u_{2,i}^b \right) z_i x_{2,-i}^a + \ldots + \left( 1 - u_{k,-i}^b \right) z_{-i} x_{k,i}^a - \left( 1 - u_{k,i}^b \right) z_i x_{k,-i}^a \right) \bar{z}^{-1} , \\ &\left( 1 - \bar{z} - \bar{x}_1 - \bar{u}_1 - \bar{x}_2 - \bar{u}_2 - \ldots - \bar{x}_k - \bar{u}_k \right) \end{aligned} \qquad (4)$$

where $x_1, x_2, \ldots, x_n$ denote the $n$ mechanisms of manipulation and $u_1, u_2, \ldots, u_n$ represent suppression. Additional terms of gain and loss from manipulation and being manipulated are added to the first parenthesis, and group productivity can potentially be decreased by all the $2k + 1$ traits.

We first examine this potentially non-equilibrium scenario by assuming all mechanisms are present from the beginning and consider the simplest case of two mechanisms ($k = 2$). By simulating various combinations of shape parameters and relatedness, we found that levels of selfishness and manipulation do not always tend to a single equilibrium. Instead, they can oscillate in a periodic style (*Figure 4a*). Although the oscillation in suppression is less profound, it also fluctuates in a smaller amplitude in response to the changes in levels of manipulation. We used harmonic regression to test for periodic oscillations, as opposed to random noise (*Figure 4b* and Appendix 1—4.2).

We found oscillations occur when relatedness is relatively low ($R < 1/6$) and when shape parameters are intermediate decelerating ($0.6 < a, b < 0.85$). In these intermediate settings, the optimal levels of manipulation and selfishness are similar, which could lead to a similar pay-off between investing in

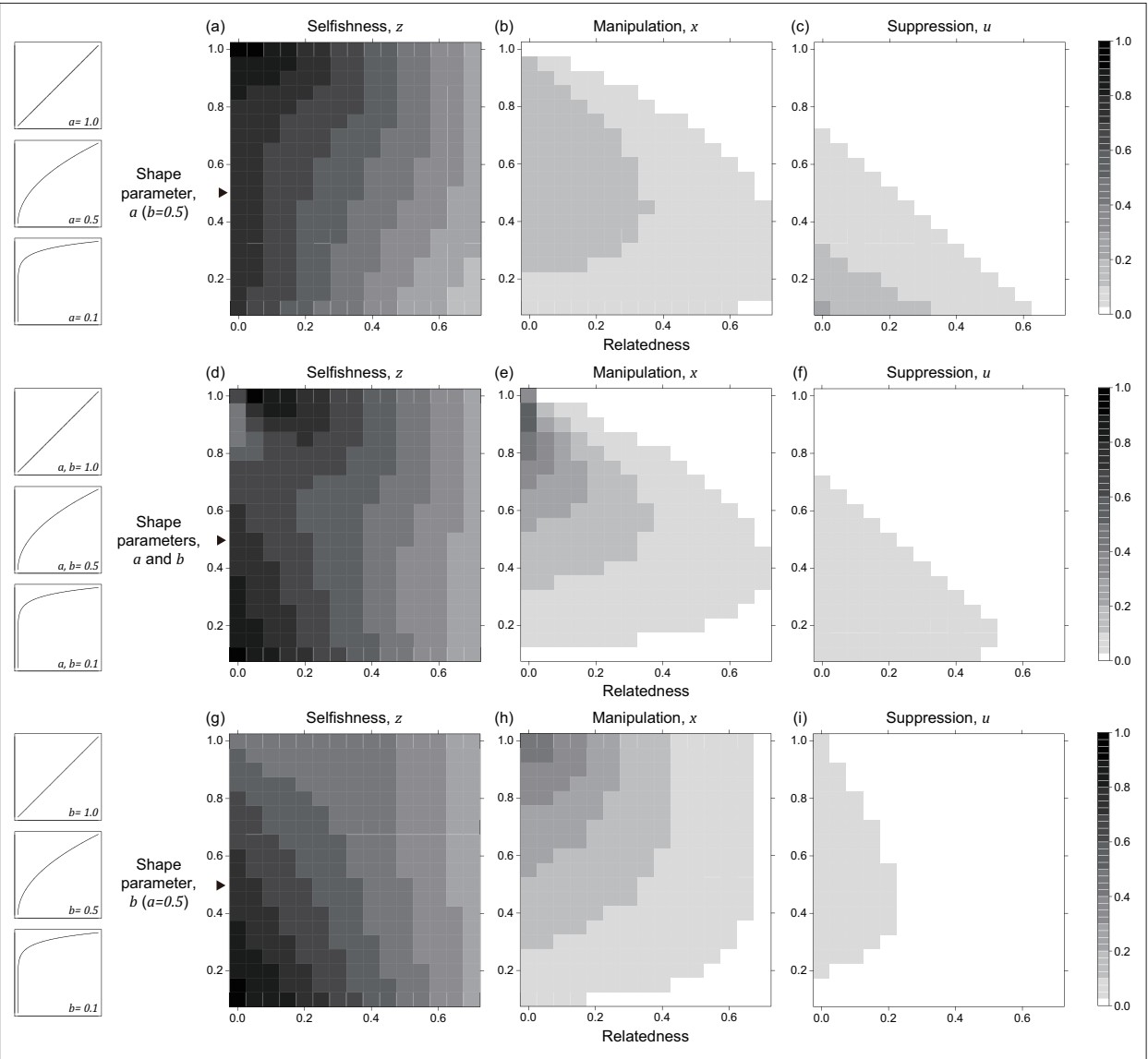

**Figure 3.** Nonlinear returns of manipulative cheating and its suppression. Panels (**a**), (**b**), and (**c**) show the predicted ESS level of selfishness ($z^*$), manipulation ($x^*$), and suppression ($u^*$) respectively, for different values of relatedness ($R$, x-axis). From top to bottom, each row of panels presents varying different parameters on y-axis: the shape parameter for manipulative cheating (a-c; $a$), both parameters (d-f; $a$ and $b$), and the shape parameter for suppression (g-i; $b$). The black triangle indicates the horizontal row where the parameter settings are identical to each other and to *Figure 2*. We have focused on decelerating function for shape parameters because it resembles the nature of communication (Appendix 1—6.1).

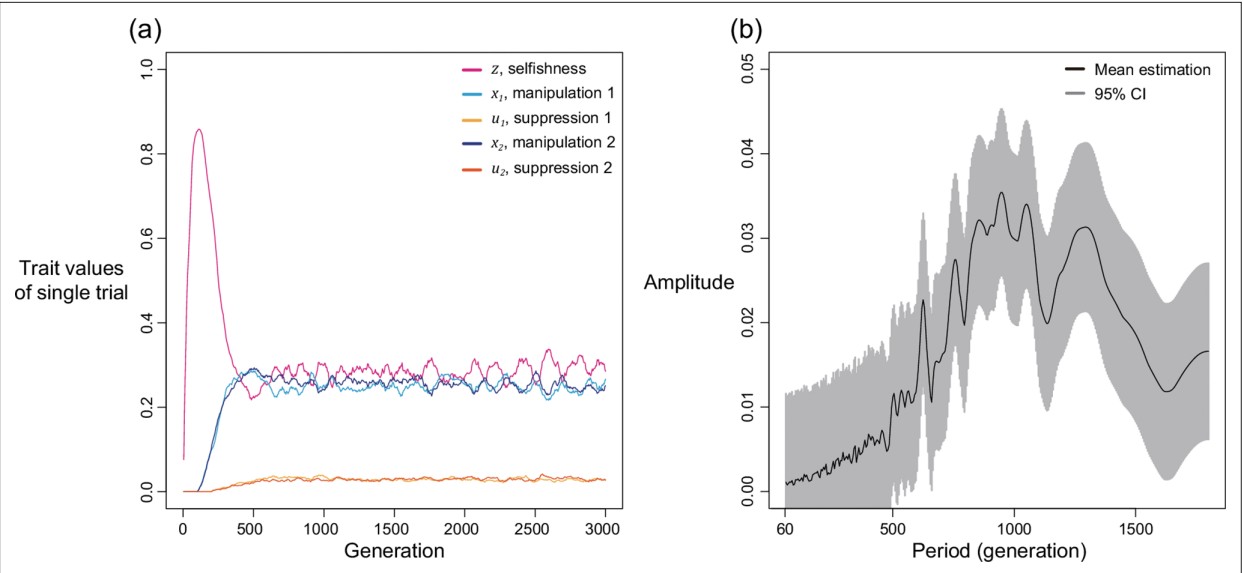

**Figure 4.** Selfishness and manipulative cheating traits can oscillate when there are two mechanisms for manipulation and suppression. (**a**) An example of oscillating dynamics when shape parameters are intermediate, and relatedness is relatively low ($R = 1/8$; $a = b = 0.7$). Broader analysis and more details are provided in Appendix 1—4. (**b**) The estimated amplitude of the time series of selfishness by harmonic regression. Harmonic regression is an analysis of time series to show the significance of periodic oscillation, where low period means frequent recurrent pattern, and high period means recurrence ager a long span. Here, 10 repeated simulations were used and only the last 75% of the 3000 generations was analysed, after selfishness is roughly stabilised. The analysis shows maximal amplitude is found at period of 815 generations with p-value of $1.8035 \times 10^{-5}$. Mean estimation is shown in black curve whereas the 95% confidence interval is illustrated as the grey area (F-test; $\mu \pm 2\sigma$).

them. Thus, any small perturbation in trait values could disturb the dynamics and generate oscillation. In support of the hypothesis that oscillation is a consequence of interactions between selfishness and manipulation, we found that oscillation can be prevented when selfishness cannot mutate (Appendix 1—4.3) or when there is no linkage between traits (Appendix 1—4.4).

We also found that oscillations can occur when there are more than two mechanisms to manipulate and supress (Appendix 1—4.5), becomes slower when mutation rate is smaller (Appendix 1—4.6), and less likely when there is a personal reproduction cost to manipulation and suppression (Appendix 1—4.7). The dynamics between cheats and cooperators can involve frequency dependent selection (*Ross-Gillespie et al., 2007*; *Gore et al., 2009*; *Patel et al., 2019*). In this model, we were examining evolution in a continuous trait space and so we were unable to test for frequency-dependent selection in the way that can be done with interactions between cheats and cooperator (i.e. 2 types, as opposed to continuous; *Allison, 2005*; *Yurtsev et al., 2013*).

We also found that recombination between traits could eliminate the oscillations (Appendix 1—4.4 and 4.8). We added recombination to our simulation in two ways. The first way is independently inheriting each trait when creating a new group founder for the next generation (Appendix 1—4.4). The second way is through allocating all founders in a founder pool and randomly grouped them into pairs. Each pair then exchange one random trait value with a recombination rate (Appendix 1—4.8). As a result, the first way is more discrete while the second way is more continuous. Because the group productivity term makes individuals with sum of all trait values exceeding 1 have no fitness, a positively deviated trait value cannot be protected by other traits simultaneously become negatively deviated under the presence of recombination. This finding coincides with previous literature that recombination prevents evolutionary branching into distinct genotypes, because it removes the linkage between strategies (linkage disequilibrium; *Abrams et al., 1993*; *Dieckmann and Doebeli, 1999*). Consequently, oscillations are more likely to be seen in asexual species.

We now consider a slightly different scenario (*Figure 5*). In the above analysis, we assumed that all manipulation and suppression mechanisms were available at the start of the simulation. An alternative possibility is that different mechanisms can arise over time, by mutation. When a manipulation trait arises, it could be selected for. Furthermore, once a form of manipulation becomes sufficiently common then a mutation that allows suppression of that manipulation could be selected for. These

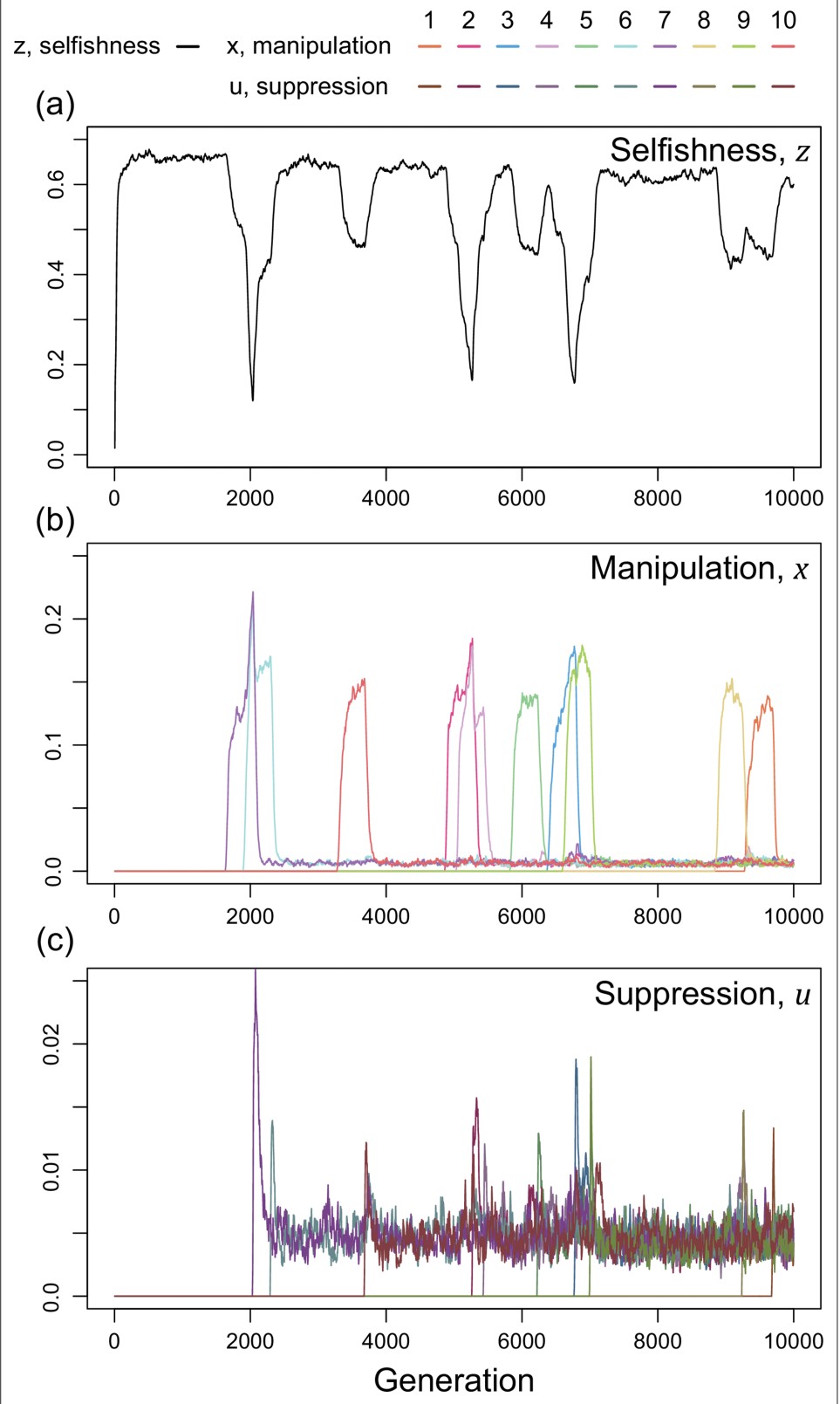

**Figure 5.** Time series of trait dynamics of the simulations when new mechanisms for manipulation and suppression are introduced at different time points. Selfishness is shown in the upper panel, the ten manipulative traits are shown in the middle panel, and the ten suppressive traits are shown in the bottom panel. The y-axis is scaled differently across the panels, to match the levels of variation. Each manipulative and suppressive trait are

*Figure 5 continued on next page*

*Figure 5 continued*

mutated to non-zero values in arbitrary time points for better visualisation. The shape parameters are set to $a = 0.5$ for manipulation and $b = 0.01$ for suppression, and the relatedness is $R = 1/3$. All data are from a single representative simulation.

different forms of manipulation could involve very different traits, or new forms of an old trait, that would require a different suppression mechanism. This scenario could potentially lead to an evolutionary arms race where new forms of manipulation evolve and are then suppressed. We modelled this scenario by introducing mutations for new manipulative traits at random time points, and then mutations for the suppression of manipulation 400 generations afterwards for illustrative purpose.

We found that this scenario also leads to a form of arms race (*Figure 5*). When a new form of manipulation arises, it initially spreads. This spread of manipulation favours lower levels of selfishness, because the manipulation reduces the benefit of selfishly investing in reproduction. In addition, once a manipulation is common, any mutation that allows suppression of that manipulation will be favoured, and hence increase in frequency. As suppression spreads, this selects for reduced investment in manipulation (which would just be blocked), and hence also increased selfish investment in reproduction (which won't be manipulated anymore) (*Figure 5*). Overall, this leads to a pattern of recurring dynamics whenever a new manipulative trait arises, and where individuals can show variable levels of selfishness, manipulation and suppression.

For illustrative purposes, our example in *Figure 5* shows the gradual evolution of 10 different manipulation mechanisms and focuses on the parameter space where suppression can almost completely block manipulation in a very effective way ($a = 0.5$, $b = 0.01$)(*Scott and West, 2019*). We examine other scenarios in Appendix 1—5 (*Appendix 1—figure 14* & *Appendix 1—figure 15*). The repeated invasion of different forms of manipulation, and their suppression, can be seen as a form of trait substitution sequence (*Champagnat, 2006*; *Méléard and Tran, 2009*). Recombination does not remove this arms race because it results from different traits spreading in turn, rather than oscillations of linked traits.

Altogether, our results from the multiple mechanisms of manipulation suggest that: (i) multiple types of manipulation can arise and / or be maintained in a population; (ii) the manipulation traits and their suppression can sometimes show oscillating dynamics, where the levels of selfishness, manipulation, and/ or suppression oscillate. The oscillation originates either from interactions between coevolving traits (endogenous; *Figure 4*) or from the introduction of new mechanisms (exogenous; *Figure 5*). These results provide possible explanations for cases where there appears to be genetic variation for the extent or form of cheating and/or cheating suppression, such as the variance between ant patrilines (*Hughes and Boomsma, 2008*), slime moulds strains (clones; *Buttery et al., 2009*; *Khare et al., 2009*; *Levin et al., 2015*), or different bacteria strains (*Kümmerli et al., 2009*; *Bruce et al., 2017*; *Butaitė et al., 2017*).

## Discussion

We found that: (1) manipulative cheating can be favoured across a wide variety of conditions; (2) manipulation is more likely to be favoured when relatedness is relatively low; (3) suppression of manipulation can also be favoured. In addition, when we allowed for multiple mechanisms (types) of manipulation, we found that non-equilibrium dynamics could occur, such as oscillations. Oscillations could be generated in either (i) a coevolutionary interaction between selfishness and manipulation; or (ii) a genetic arms race between different forms of manipulative cheating and their specific suppressors. These non-equilibrium dynamics could help explain the genetic diversity that has been observed for both cheating and the response to cheating across the biological world (*Hughes et al., 2003*; *Linksvayer, 2006*; *Anderson et al., 2008*; *Hughes and Boomsma, 2008*; *Buttery et al., 2009*; *Khare et al., 2009*; *Kümmerli et al., 2009*; *Mitchell et al., 2012*; *Stürup et al., 2014*; *Levin et al., 2015*; *Ostrowski et al., 2015*; *Pollak et al., 2016*; *Bruce et al., 2017*; *Butaitė et al., 2017*; *Noh et al., 2018*; *Ostrowski, 2019*; *Furubayashi et al., 2020*; *Bruce et al., 2021*; *Mizuuchi et al., 2022*).

Our general aim was to investigate scenarios where cheating is characterised by more than just 'cooperate less'. We did this by developing a theoretical model where individuals could manipulate others, to make them cooperate more. There is a range of other possibilities for more complex forms

of cheating, such as obtaining a disproportionate share of some public good, where we suggest that a similar logic would apply (i.e. the evolution of 'more efficient' cheating). A key distinction here is that we are not modelling some evolutionarily stable signalling system which benefits the signaller and receiver (*Maynard Smith and Harper, 2003*). We have instead modelled coercion, which provides no benefit to the recipient of that coercion. Consequently, the response would be to evolve to suppress or ignore that coercion, rather than evolve to some evolutionarily stable mutually beneficial response. It would be useful to examine the consequences of different genetic systems, especially those with asymmetric inheritance such as haplodiploidy. Another useful extension would be to examine caste determination mechanism in social insect, and whether predictions depended upon whether castes are determined by environmental or genetic cues (*Schwander et al., 2010*).

Manipulative cheating could take many different forms in different species. In bacteria and other microorganisms, individuals could produce coercive molecules that make other cells invest more into the production of public goods, and invest less into reproduction. In species that form fruiting bodies, manipulation could cause the cells of other lineages to be more likely to become stalk cells (*Buttery et al., 2009*; *Khare et al., 2009*; *Levin et al., 2015*). In organisms such as social insects, the method of manipulation would depend upon how caste is determined (*Schwander et al., 2010*). In species where workers control the caste of offspring, manipulation could cause workers from other lineages (different queens or patrilines) to be less likely to rear their own offspring as reproductive, and be more likely to rear the offspring of the manipulating lineage as reproductive (*Hamilton, 1972*). In species where larvae control their own caste, manipulation could decrease the likelihood that other individuals develop as reproductive (*Wenseleers and Ratnieks, 2004*). Alternatively, manipulation could occur through more direct approaches, such as selectively killing the eggs from other lineages. These mechanisms all share the same concept that the 'more manipulative' lineage gains a greater share of group productivity.

Similarly, suppression could involve a variety of different mechanisms, from evolving to ignore coercion, to blocking behaviours that steal resources or kill individuals, to making fewer public goods. Experiments on slime moulds have found that strains can evolve to become more resistant to cheats (*Levin et al., 2015*). Bacteria appear to be able to produce more specific and less exploitable public goods (*Kümmerli et al., 2009*; *Bruce et al., 2017*; *Butaitè et al., 2017*). Different lineages of cooperative RNA replicators can survive better against different cheating replicators (*Mizuuchi et al., 2022*). These cases share the same concept that cooperators can somewhat defend against or suppress the cheats. Other possible mechanisms are discussed elsewhere, for bacteria and viruses.

Previous theoretical work has focused on conflict suppression and hence how the tragedy of the commons could be avoided (*Frank, 1995*; *West et al., 2002*; *Frank, 2003*; *Wenseleers et al., 2003*; *Wenseleers et al., 2004*; *Ratnieks et al., 2006*). In these previous papers, selfishness could be selected against, by mechanisms such as policing, sanctions, and social reputation, which alter the costs and benefits of behaving selfishly (*Frank, 1995*; *Milinski et al., 2002*; *Dionisio and Gordo, 2006*; *Dionisio and Gordo, 2007*; *El Mouden et al., 2010*). In contrast, we have examined a different route to increased selfish behaviour – the manipulation of others via manipulative cheating. Furthermore, we have shown that if there are multiple manipulation traits, then genotypic diversity can be maintained for selfishness, manipulative cheating, and/or its suppression. This diversity result links to previous work on quorum sensing in bacteria, which has shown how cycles of 'cheating' and 'cheating immunity' can arise (*Eldar, 2011*; *Pollak et al., 2016*). Several theoretical models have examined the coevolution of offense and defence traits in ecological interactions (*Press and Dyson, 2012*; *Hilbe et al., 2013*) and in sexual conflict (*Gavrilets, 2000*; *Gavrilets et al., 2001*; *Rowe et al., 2005*). These models examined specific ecological scenarios, such as memory of previous interactions or runaway processes. Other recent models have examined a different manipulation question- how individuals can be favoured to manipulate relatives when this leads to an indirect fitness benefit to the manipulator, and the manipulated individual is not selected to resist (*González-Forero and Gavrilets, 2013*; *González-Forero, 2014*; *González-Forero, 2015*; *González-Forero and Peña, 2021*). Manipulation can also be favoured in mutualisms if it leads to better terms of trading (*Wyatt et al., 2016*). More generally, manipulation and counteradaptations to prevent elimination can arise in many forms of communication (*Dawkins et al., 1979*; *Maynard Smith and Harper, 2003*).

To conclude, we have shown how cheating can take more complex forms than just lower levels of cooperation. We have shown that individuals can also be selected to manipulate the behaviour

of others, for their own selfish gain. Furthermore, this can lead to evolutionary arms races between attempts to manipulate and to suppress that manipulation. A major task for the future is to determine the extent to which more complex forms of cheating occur in the natural world (*Pollak et al., 2016*; *Meir et al., 2020*; *Leeks et al., 2021*). In many cases, this would require elucidation of the underlying mechanisms, which could be completely different across different species. The potential for manipulation and its suppression could arise at all levels of life, from RNA replicators to complex animal societies.

## Acknowledgements

We thank Thomas W Scott, Guy A Cooper and the rest of West group for the input since the early development stages of this project; for discussions with Matishalin Patel, Xiang-Yi Li Richter, and Daisuke Kyogoku, who provided us with valuable literature. We are grateful for the three anonymous reviewers for their helpful feedback.

## Additional information

### Funding

| Funder | Grant reference number | Author |
|---|---|---|
| European Research Council | Horizon 2020 Advanced Grant 834164 | Stuart Andrew West Ming Liu |
| Ministry of Education | Oxford-Taiwan Graduate Scholarships | Ming Liu |

The funders had no role in study design, data collection and interpretation, or the decision to submit the work for publication.

### Author contributions

Ming Liu, Conceptualization, Formal analysis, Investigation, Visualization, Writing – original draft, Writing – review and editing; Stuart Andrew West, Conceptualization, Supervision, Funding acquisition, Investigation, Writing – original draft, Project administration, Writing – review and editing; Geoff Wild, Conceptualization, Formal analysis, Validation, Investigation, Writing – original draft, Writing – review and editing

### Author ORCIDs

Ming Liu ⓘ http://orcid.org/0000-0002-5170-8688
Stuart Andrew West ⓘ http://orcid.org/0000-0003-2152-3153
Geoff Wild ⓘ http://orcid.org/0000-0001-7821-7304

### Decision letter and Author response

Decision letter https://doi.org/10.7554/eLife.80611.sa1
Author response https://doi.org/10.7554/eLife.80611.sa2

## Additional files

### Supplementary files

• MDAR checklist

### Data availability

All results are generated using C and Python. The codes and data used for this study are available at: https://github.com/mingpapilio/Codes_Manipulative_Cheat, (copy archived at swh:1:rev:e526dcaff01d51c334b90977cf17793c2c255e67).

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

## Appendix 1

Our appendix contains seven sections. In section 1, we formally analyse our model with single mechanisms to manipulate and suppress manipulation. Section 2 contains the non-suppression model, which is a reduced model without suppressive traits. This model bridges the gap between the single-mechanism model and the classic tragedy of the commons (*Frank, 1994*; *Frank, 1996*; *Taylor and Frank, 1996*; *Frank, 1998*; *Dionisio and Gordo, 2006*; *Frank, 2010a*; *Frank, 2010b*). Further, the non-suppression model also illustrates a biologically possible case where manipulation cannot be suppressed. We consider the effects of personal cost in manipulation and suppression in section 3. Section 4 provides additional analyses for the scenario with two mechanisms to manipulate and suppress, where we found oscillations in selfishness and manipulation (*Figure 4*). We examine the origin of the oscillation dynamics and compare the results with game-theoretical analysis. We subsequently present similar arms race dynamics in section 5 for the scenario with multiple mechanisms introduced at different time points (*Figure 5*), with different biological settings assumed in each case. In section 6, we discuss some modelling details and justification. Lastly, section 7 has the details for the individual-based simulations.

# 1. Formal analysis of the scenario with a single mechanism to manipulate and suppress manipulation

## 1.1 Derivation of the fitness equation

We track a population of haploid asexual individuals by observing it at discrete points spaced evenly in time. The model population itself is subdivided into groups of size $n$. Groups are re-constituted at the very end of a given time step, through among group competition. Individuals compete within their own group for their share of the group's productivity.

We use $y_i$ to denote the (absolute) competitive effort realised by focal individual $i$. We build up the fitness by assuming that the focal individual initially commits to a within-group level of competition equal to $z_i$. Likewise, each non-focal individual $k \neq i$ commits to a within-group level of competitive effort equal to $z_k$.

Once each group member has committed to an initial level of competitive effort, the group enters a phase in which members can manipulate effort (i.e., investment) from one another. We model the manipulation phase by assuming that the focal group member is assigned a non-self partner, uniformly at random with replacement. The assumption is equivalent to letting the focal individual manipulate every neighbour to the same degree and we use a probabilistic sampling narrative for the ease of explaining each process (**Wild, 2011**). When manipulation takes place, the focal individual attempts to manipulate a fraction $x_i^a$ of the competitive effort $z_k$ committed to by the partner. Here, we are treating $x_i$ as a new trait (i.e., effort devoted to manipulation), and $a$ as a constant that modulates the efficacy of the trait (i.e., determines how effective the effort is). The partner individual can also suppress manipulation. Specifically, we assume that the partner individual prevents a fraction $(1 - u_k^b)$ of manipulation that would have otherwise occurred. Here, we are thinking of $u_k$ as a trait, and $b$ as a constant that modulates the efficacy of the trait.

The focal individual is not only a manipulator, it can also be a the target of manipulation. In keeping with the description above, the focal individual is chosen to be manipulated by each of its non-self neighbours with probability $1/(n-1)$. When chosen, the focal individual loses $x_k^a \left(1 - u_i^b\right)$ of the competitive effort $z_i$ committed to by itself.

Putting the details together, we arrive at the following equation for the competitive effort realised by focal individual $i$:

$$
\begin{aligned}
y_i &= z_i + x_i^a \frac{1}{n-1} \sum_{k \neq i} \left(1 - u_k^b\right) z_k - \left(1 - u_i^b\right) z_i \frac{1}{n-1} \sum_{k \neq i} x_k \\
&= z_i + x_i^a \left(1 - \bar{u}_{-i}^b\right) \bar{z}_{-i} - \bar{x}_{-i} \left(1 - u_i^b\right) z_i
\end{aligned}
\tag{A1}
$$

where a bar paired with a $-i$ subscript denotes the arithmetic mean of non-focal individuals in the focal group. It is important to note that the share of group productivity of the average non-focal individual can be expressed as

$$
\begin{aligned}
\bar{y}_{-i} &= \frac{1}{n-1} \sum_{k \neq i} \left( z_k + x_k^a \frac{1}{n-1} \left(1 - u_i^b\right) z_i - \frac{1}{n-1} x_i^a \left(1 - u_k^b\right) z_k + x_k^a \frac{1}{n-1} \sum_{j \neq i,k} \left(1 - u_j^b\right) z_j \right. \\
&\quad \left. - \frac{1}{n-1} \left(1 - u_k^b\right) z_k \sum_{j \neq i,k} x_j^a \right) \\
&= \bar{z}_{-i} + \frac{1}{n-1} \bar{x}_{-i} \left(1 - u_i^b\right) z_i - \frac{1}{n-1} x_i^a \left(1 - \bar{u}_{-i}^b\right) \bar{z}_{-i} + \frac{1}{(n-1)^2} \left( \sum_{k \neq i} \sum_{j \neq i,k} x_k^a \left(1 - u_j^b\right) z_j \right. \\
&\quad \left. - \sum_{k \neq i, j \neq i,k} x_j^a \left(1 - u_k^b\right) z_k \right)
\end{aligned}
$$

Further simplification can be achieved by recognising that the term contained in parenthesis immediately above is zero. To see this let $\delta_{kj} = \delta_{jk}$ take a value of 1 if $k = j$ and a value of 0 otherwise. Then,

$$\sum_{k \neq i, j \neq i,k} \sum x_k^a \left(1 - u_j^b\right) z_j - \sum_{k \neq i, j \neq i,k} \sum \left(1 - u_k^b\right) z_k x_j^a =$$

$$\sum_{k,j \neq i} x_k^a \left(1 - u_j^b\right) z_j \left(1 - \delta_{kj}\right) - \sum_{k,j \neq i} x_j^a \left(1 - u_k^b\right) z_k \left(1 - \delta_{jk}\right) = 0 \tag{A3}$$

In words, the term in question simplifies to zero because the double sums both represent the total sum of all entries of the same $(n - 1) \times (n - 1)$ matrix. Therefore, the average competitive effort in the focal group is:

$$\begin{aligned}
\bar{y} &= \tfrac{1}{n} y_i + \tfrac{n-1}{n} \bar{y}_{-i} \\
&= \tfrac{1}{n} z_i + \tfrac{n-1}{n} \bar{z}_{-i} + \tfrac{1}{n} x_i^a \left(1 - \bar{u}_{-i}^b\right) \bar{z}_{-i} - \tfrac{n-1}{n} \tfrac{1}{n-1} x_i^a \left(1 - \bar{u}_{-i}^b\right) \bar{z}_{-i} \\
&\quad - \tfrac{1}{n} \bar{x}_{-i} \left(1 - u_i^b\right) z_i + \tfrac{n-1}{n} \tfrac{1}{n-1} \bar{x}_{-i} \left(1 - u_i^b\right) z_i \\
&= \bar{z}
\end{aligned} \tag{A4}$$

Where bars with no subscript denote an arithmetic mean taken over the entire group of $n$ individuals. The average competitive effort in the group is not changed by the manipulation phase. Manipulation only serves to redistribute the initial commitment to competitive effort within a given group. It follows that the relative share of group productivity of the focal individual can be expressed as $y_i / \bar{z}$ .

We next assume, for simplicity, that all traits impose a cost on group performance at the same rate, and we propose that group productivity is proportional to $1 - \bar{z} - \bar{x} - \bar{u}$ . Thus, we express fitness of the focal individual as

$$w_i = \frac{y_i}{\bar{z}} \left(1 - \bar{z} - \bar{x} - \bar{u}\right) \tag{A5}$$

Technically, $w_i$ in the previous line is more accurately described as an inclusive-fitness-generating function, because differentiating it following the steps outline in *Taylor and Frank, 1996* will lead to the correct generalisation of Hamilton's rule (*Hamilton, 1964a*; *Hamilton, 1964b*).

## 1.2 Kin selection analysis

Following *Taylor and Frank, 1996* , we assume that traits $z$, $x$, and $u$ are controlled by a separate locus. We use, $g_z$ , $g_x$ , and $g_u$ , respectively, to denote the genotypic value of the focal individual at the $z$, $x$, and $u$ loci.

To capture the effect of competition on an individual's inclusive fitness we treat the traits $z_i$ , $\bar{z}_{-i}$ , and $\bar{z}$ as a function of $g_z$ . We then differentiate $w_i$ with respect to $g_z$ and evaluate the result at a point where all individuals in the population have the same genotypic value at every locus. Evaluating at such a point allows us to assert that $z_i = \bar{z}_{-i} = \bar{z} = z$, $x_i = \bar{x}_{-i} = \bar{x} = x$, and $u_i = \bar{u}_{-i} = \bar{u} = u$. It also justifies us replacing $dz_i / dg_z$ with relatedness between the focal individual and itself (= 1), replacing $d\bar{z}_{-i} / dg_z$ with relatedness between the focal individual and its average non-self group mate (= $\bar{R}_{-i,z}$, where the subscript reminds us that relatedness is assessed at the $z$ locus), and replacing $d\bar{z} / dg_z$ with relatedness between the focal individual and the average member of its group, self-included (= $\bar{R}_z$). Following through this entire multi-step procedure, we arrive at

$$\Delta W_z = \left( \bar{R}_{-i,z} \frac{-\left(\frac{n-1}{n} - x^a \left(1 - u^b\right)\right)}{z} + \frac{\frac{n-1}{n} - x^a \left(1 - u^b\right)}{z} \right) \left(1 - z - x - u\right) - \bar{R}_z \tag{A6}$$

as the inclusive fitness effect of a small increase in selfishness, $z$. When $\Delta W_z$ is zero, $z$ is at equilibrium with respect to the action of selection (ESS).

Treating traits $x_i$ , $\bar{x}_{-i}$ , and $\bar{x}$ as a function of $g_x$ we can go through analogous steps to obtain

$$\Delta W_x = \left( \bar{R}_{-i,x} \left(-ax^{a-1}\right) \left(1 - u^b\right) + ax^{a-1} \left(1 - u^b\right) \right) \left(1 - z - x - u\right) - \bar{R}_x \tag{A7}$$

where the subscript reminds us that relatedness is now assessed at the $x$ locus. Treating traits $u_i$ , $\bar{u}_{-i}$ , and $\bar{u}$ as a function of $g_u$ , we get

$$\Delta W_u = \left( \bar{R}_{-i,u} \left( -x^a \right) bu^{b-1} + x^a bu^{b-1} \right) \left( 1 - z - x - u \right) - \bar{R}_u \tag{A8}$$

as the inclusive fitness effect of suppression, where relatedness is assessed at the $u$ locus.

## 1.3 Mathematical criterion of manipulation in other theoretical papers

We compare our model with other theoretical papers, which examine related concepts. We start with Frank's (1995) policing model:

$$w_i = \left( \bar{a} - ca_i + \left( 1 - \bar{a} \right) z_i / \bar{z} \right) \left( 1 - \left( 1 - \bar{a} \right) \bar{z} \right), \tag{A9}$$

where $a_i$ and $z_i$ are the degree of policing and selfishness of the focal individual, $\bar{a}$ and $\bar{z}$ are the averages in the focal group. $c$ is the cost of policing. Policing changes the benefit and cost of selfishness for all group members, with each multiplied by $1 - \bar{a}$ . In contrast, our manipulative cheating involves the actor coercing the recipient into being less selfish (more cooperative).

*González-Forero and Gavrilets, 2013* examined a different form of 'manipulation'. In their model, fitness is depending on the behaviour of three types: manipulator that performs manipulation, recipient of manipulation, and a benefactor that is not involved in manipulation but benefit from its social partners' behaviour:

$$w = 1 - pc_{manipulate}k_{manipulator} - \left( c_{acquiesces}P \left( 1 - q \right) - c_{resistance}Pq \right) k_{recipient} + b\Pi \left( 1 - Q \right) k_{benefactor} , \tag{A10}$$

where $k$ are the reproductive values, $c$ are the cost for taking each option, $p$ is the probability for a manipulator to act on a reachable subject, $q$ is the probability for a subject to resist, $P$ is the average manipulation probability, $Q$ is the average resistance probability, and $\Pi$ is the average manipulation probability that manipulation occurs and successfully reaches a recipient. In addition to the more complicated pay-off setting, their model assumes that social interactions are tiggered by manipulation. They are examining when an manipulator can be favoured to manipulate a related recipient, to behave in a way that provides an indirect benefit to the recipient, and that this benefit is sufficient that the recipient is not favoured to resist the manipulator. This contrasts with our model, where manipulative cheating involves manipulation of selfishness (cooperation) – manipulation of an already occurring social behaviour to provide a selfish benefit. Therefore, although both models use the term 'manipulation', they are modelling very different biological scenarios.

## 2. The non-suppression model: model with only selfishness and one mechanism for manipulation

We assume suppression is absent in this model, but manipulation can occur. Consequently, this gives:

$$w_i = \frac{z_i + z_{-i}x_i^a - z_i x_{-i}^a}{\bar{z}}\left(1 - \bar{z} - \bar{x}\right), \tag{A11}$$

where $x_i$ is the level of manipulative cheating of the focal individual, $z_{-i}$ is the average level of selfishness (excluding the focal individual), $a$ is the shape parameter for manipulative cheating, and $1 - \bar{z} - \bar{x}$ is the group's productivity. As above, the parameter $a$ determines whether the benefit from increased investment into manipulation is decelerating ($a < 1$), linear ($a = 1$) or accelerating ($a > 1$).

To facilitate comparison with previous work, the relations between the classic tragedy of the commons, the non-suppression model, and the model with suppression are illustrated in **Appendix 1—figure 1**. The first model is the classic tragedy of the commons model, which predicts a single optimal level of investment into cooperation / selfish reproduction (**Appendix 1—figure 1a**; *Frank, 2010a*). The next model is the non-suppression model, which allows for manipulative cheating, where individuals increase their own investment into reproduction, by reducing that of others (**Appendix 1—figure 1b**). Lastly, in the model with suppression, we allow suppression as a defensive mechanism against manipulative cheating (**Appendix 1—figure 1c**).

(a) Classic tragedy of the commons

(b) The non-suppression model: tragedy+ manipulation

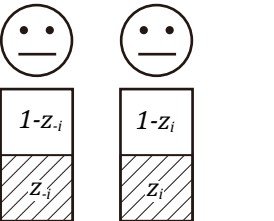 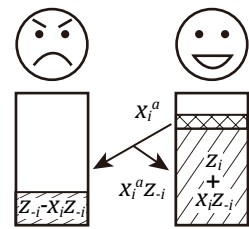

(c) The model with suppression (main model): tragedy+ manipulation+ suppression

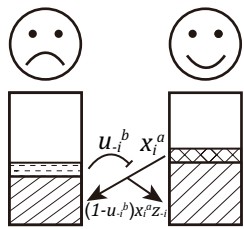

|  | Classic tragedy of the commons | Non-suppression model | The model with suppression |
|---|---|---|---|
| Group productivity | $1 - \bar{z}$ | $1 - \bar{z} - \bar{x}$ | $1 - \bar{z} - \bar{x} - \bar{u}$ |
| Individual share of group productivity | $\dfrac{z_i}{\bar{z}}$ | $\dfrac{z_i + x_i^a z_{-i} - x_{-i}^a z_i}{\bar{z}}$ | $\dfrac{(z_i + (1-u_{-i}^b)x_i^a z_{-i} - (1-u_i^b)x_{-i}^a z_i)}{\bar{z}}$ |

**Appendix 1—figure 1.** Comparison between the classic tragedy of the commons, the non-suppression model, and the model with suppression. Here, we illustrate three models with gradually increasing complexity. (**a**) In the classic tragedy of the commons, individuals can choose to invest resources into reproduction ($z_i$) or group productivity (cooperation; $1 - \bar{z}$). (**b**) In the non-suppression model, individuals can also manipulate others ($x_i$), to make them invest more in group productivity. (**c**) In the model with suppression, individuals can also suppress (block) the manipulation by others ($u_i$). Within each model, arrows indicate manipulation, the blockings indicate suppression, and the outcome of the two actions are expressed below the arrows. The fitness of an individual is divided into two components: group productivity and share of group productivity, both shown at the bottom of the figure.

We find the change of inclusive fitness effect for selfishness, and manipulation:

$$\begin{cases} \Delta W_z = \left( \bar{R}_{-i,z} \frac{-\left(\frac{n-1}{n} - x^a\right)}{z} + \frac{\frac{n-1}{n} - x^a}{z} \right) (1 - z - x - u) - \bar{R}_z \\ \Delta W_x = \left( \bar{R}_{-i,x} \left( -ax^{a-1} \right) + ax^{a-1} \right) (1 - z - x - u) - \bar{R}_x \end{cases}.$$

(A12)

*Equation A12* could not be solved analytically, and so we found solutions with an iterative numerical analysis. We also tested the predictions of our analytical model with an individual-based simulation, which showed close agreement (*Appendix 1—figure 2b*).

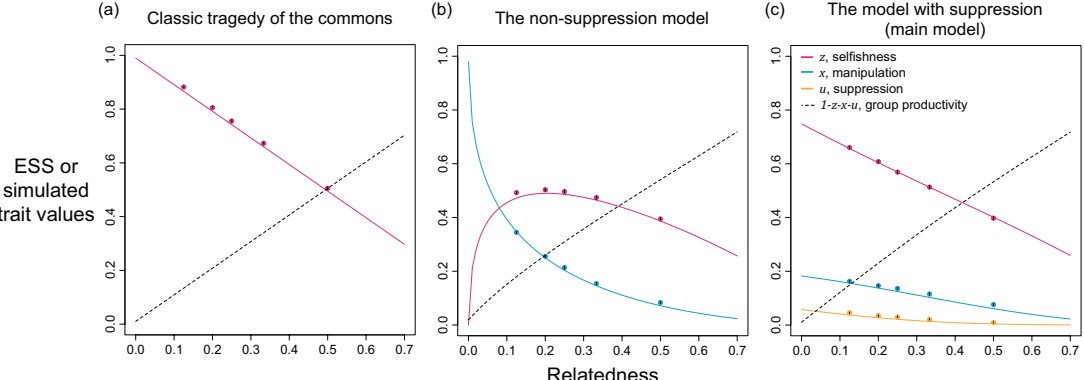

**Appendix 1—figure 2.** Selfishness, manipulation, and suppression of manipulation. The panels show the ESS predictions and simulation results for: (**a**) tragedy of the commons (only selfishness evolves); (**b**) the non-suppression model (selfishness and manipulation evolve); (**c**) the model with suppression (selfishness, manipulation, and suppression of manipulation evolve). The solid lines show the ESS predictions, and the dots show the results of our individual-based simulations ($a = b$=0.5).

The non-suppression model predicts that manipulation can be favoured, and that greater levels of manipulation will be favoured when relatedness is lower (*Appendix 1—figure 2b*). This prediction arises because there is an indirect fitness cost to manipulating relatives and reducing group productivity. Further, since a function with smaller shape parameter means the manipulative effective is higher for a given trait value, as shown in the three panels on the left of *Appendix 1—figure 2*, we find greater intermediate values of manipulation when the shape is more decelerating (smaller $a$ has more shaded area, *Appendix 1—figure 3b*).

This model also predicts a very different relationship between selfishness and relatedness, compared with the classic tragedy of the commons model and the model with suppression (compare *Appendix 1—figure 2b* with other panels). Specifically, rather than selfishness increasing as relatedness decreases, the non-suppression model predicts that selfishness shows a non-monotonic (domed) pattern, peaking at intermediate relatedness (*Appendix 1—figure 2b*). The difference in the non-suppression model is that, relative to tragedy of the commons, selfishness is not as advantageous when relatedness is low. At low relatedness, there is selection for high levels of manipulation – this dilutes the benefit of selfish reproduction because that effort will be taken away by manipulation. Within our lottery ticket metaphor in the main text, the benefit of buying tickets is reduced if those tickets are just going to be stolen.

Comparing the non-suppression model and the model with suppression, a consequence of adding in suppression is that lower levels of manipulation are selected for, because the benefit of increased manipulation is reduced if it will be suppressed (compare *Appendix 1—figure 2b & c*). This lower level of manipulation, and the fact that some of it is suppressed, lead to selfishness being beneficial at low relatedness, as in the classic tragedy of the commons model (compare *Appendix 1—figure 2a & c*), and in contrast to when there is no suppression (*Appendix 1—figure 2b*).

By comparing all three models, we also found the group productivity did not vary depending upon whether or not manipulation and suppression were included in our model, compared to the classic tragedy of the commons model (*Appendix 1—figure 2*). This is because adding manipulation and suppression does not change the basic structure of the model: share of productivity multiplied

by group productivity. Thus, the relation between relatedness and group productivity is not affected: the two additional traits are sharing the 'loss' in group productivity with selfishness, where the sum of loss is the same for a given relatedness. In other words, manipulation and suppression do not change the degree of cooperation across the group and this finding confirms the design in *equation A4*.

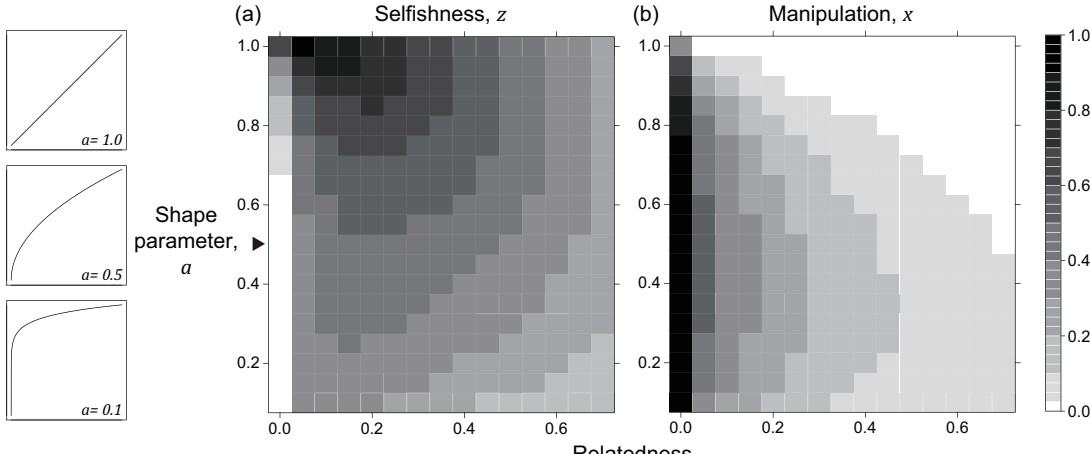

**Appendix 1—figure 3.** Nonlinear returns in the non-suppression model. Panels (**a**) and (**b**) show the predicted ESS level of selfishness ($z^*$) and manipulation ($x^*$) respectively, for different values of relatedness ($R$) and the shape parameter ($a$). The black triangle indicates the horizontal row where the parameter settings are identical to each other and to *Appendix 1—figure 2b*.

# 3. The models with personal costs: adding cost to relative shares of group productivity

In the main models, investment in manipulation or suppression led to reduced investment in group productivity. As an alternative, we also investigated a scenario where manipulation and selfishness have a personal cost, reducing the share that an individual obtains of the group productivity, but not influencing the overall success of the group. In this case, manipulative cheating and suppression of cheating now have impacts on the average level of competition, and they are not just redistributing the share of productivity. We first consider the case where suppression is absent and manipulation has a personal cost, and then we move on to the case where both manipulation and suppression are allowed.

## 3.1 The non-suppression model with personal costs

The personal cost of manipulation can be added in many ways, and one easy way is to multiply the benefit of selfishness by $(1 - x)$, where $x$ is the level of manipulation. For simplicity, we assume cost is multiplicative and there is no coefficient before $x$. This modification changes the competitive effort through selfishness from $z_i$ to $(1 - x_i) z_i$, and changes the competitive effort gained through manipulation from $z_{-i} x_i^a$ to $(1 - x_{-i}) z_{-i} x_i^a$. Similarly, the average effort of the group is $(1 - \bar{x}) \bar{z}$, making the fitness of the focal individual,

$$w_i = \frac{(1-x_i)z_i+(1-x_{-i})z_{-i}x_i^a-(1-x_i)z_ix_{-i}^a}{(1-\bar{x})\bar{z}} \left(1 - \bar{z} - \bar{x}\right), \tag{A13}$$

where $\left(1 - \bar{z} - \bar{x}\right)$ is the group productivity and the terms before it is the relative share of group productivity. By comparing *equations A13* and *A11*, it is apparent that we are multiplying $(1 - x)$ for both the numerator and denominator in *equation A11*. However, it is meaningful because each term is multiplied by different $x$: some are the level of manipulation of the focal individual, while others are the average of the social neighbours or the average across the entire group.

We found that adding a personal cost of manipulation leads to a reduced level of manipulation, especially when $R < 0.1$ (compare *Appendix 1—figure Appendix 1—figures 2b and 4b*). Furthermore, because both selfishness and manipulation have a personal cost, the net benefit obtained from investing in manipulation is reduced, and so we found no dramatic increase in the predicted level of manipulation when $R < 0.1$ (*Appendix 1—figure 4b*). In contrast, when manipulation is cost-free at the individual level, the predicted strategy is to put all selfish effort into manipulation when relatedness of the population is very low (*Appendix 1—figure 2b*).

## 3.2 The model with suppression and personal cost

We then allow suppression, with a personal cost, that reduces the share of group productivity obtained. We multiply each term in the relative share of productivity in *Equation 1* with $(1 - x - u)$, where $x$ and $u$ are the ones with same identity as $z$ they are multiplying e.g., $z_i$ multiplied by $(1 - x_i - u_i)$. This gives:

$$w_i = \frac{(1-x_i-u_i)z_i+\left(1-u_{-i}^b\right)(1-x_{-i}-u_{-i})z_{-i}x_i^a-\left(1-u_i^b\right)(1-x_i-u_i)z_ix_{-i}^a}{(1-\bar{x}-\bar{u})\bar{z}} \left(1 - \bar{z} - \bar{x} - \bar{u}\right). \tag{A14}$$

We found the predicted level of suppression is lower when a personal cost of suppression is also included to the model (compare *Appendix 1—figure 2c* & *Appendix 1—figure 4c*). Because there is (i) a lower need for suppression from a lower predicted level of manipulation, and (ii) a more costly design of suppression, we only find small levels of suppression when relatedness is low (*Appendix 1—figure 4c*). In addition, given the presence of suppression, the predicted level of manipulation is lower than the model without suppression (compare *Appendix 1—figure 4b* & *c*). Despite the lower predicted ESS values, these models with personal cost for manipulation and suppression still predict both traits can be favoured, especially when relatedness is low.

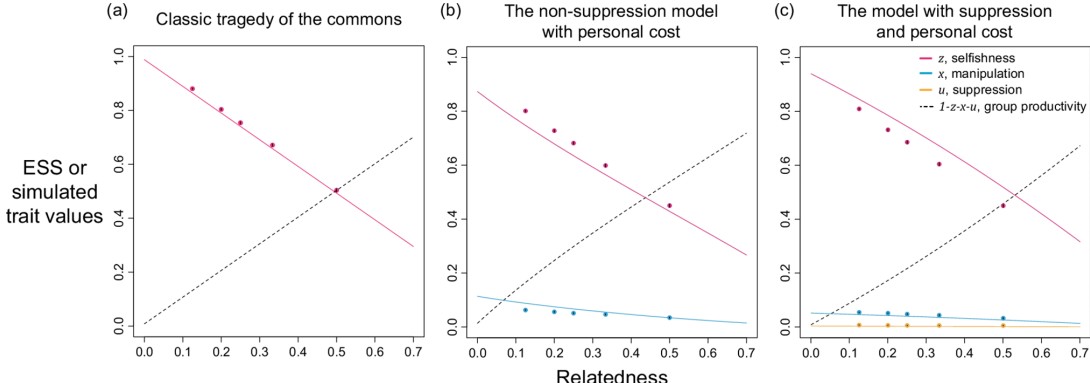

**Appendix 1—figure 4.** Selfishness, manipulation, and suppression of manipulation. The panels show the ESS predictions and simulation results for: (**a**) tragedy of the commons (only selfishness evolves); (**b**) the non-suppression model with personal cost (both selfishness and manipulation evolve); (**c**) the model with suppression and personal cost (selfishness, manipulation, and suppression evolve). All results are generated with the same shape parameters as other figures ($a = b = 0.5$).

## 4. Additional analysis of the scenario with two mechanisms where all traits are simultaneously introduced

### 4.1 The relation between ESS and relatedness

In order to understand the oscillating dynamics (*Figure 4*) better, we perform game-theoretical analysis for the two-mechanisms case and examine when does the game theoretical analysis disagree with simulations. We begin by analysing the model along relatedness and found it gives very similar predictions to the scenario with only one mechanism to manipulate (compare *Figure 2* & *Appendix 1—figure 5*). Selfishness, manipulation, and suppression are all greater when relatedness is lower. Both the two manipulative traits, and the two suppression traits converge to the same ESS values. Notably, because of the constraints on group productivity, where the sum of traits cannot exceed 1, a lower level of selfishness is selected in the scenario with two mechanisms while the levels of manipulation and suppression is roughly the same between the scenario with single mechanism and two mechanisms. In other words, because the additional manipulative mechanism is the same worthy to invest as the other one, we find a lower level of selfishness in the scenario with two mechanisms.

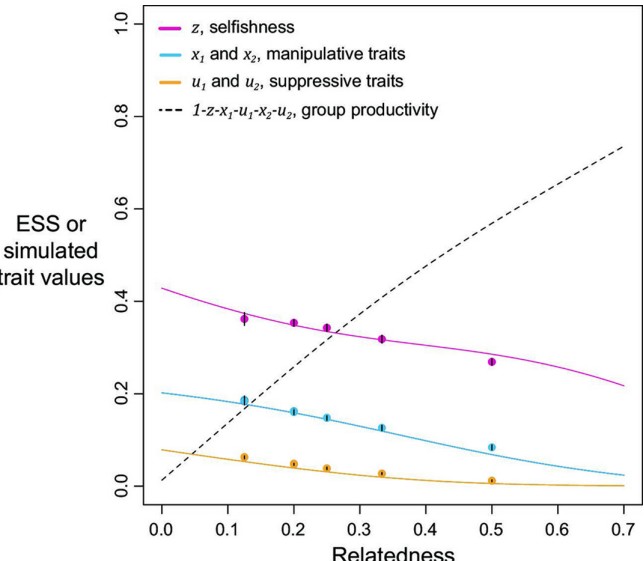

**Appendix 1—figure 5.** Selfishness, manipulations, and suppressions of manipulation in the scenario with two mechanisms to manipulate and suppress manipulation, where all traits are introduced at the same time point. Solid lines show the ESS predictions, and dots show the results of our individual-based simulations ($a = b = 0.5$).

### 4.2 The occurrence of oscillation is related to shape parameters

In contrast to the close agreement shown above, we found some disagreement between simulation and game-theoretical analyses when we analyse along shape parameters (*Appendix 1—figure 6*). We used harmonic regression to test for oscillating dynamics in our simulation data. In a harmonic regression, white noise (randomly time series) would be a flat line because all frequencies have similar predicted amplitude; but an oscillating pattern would lead to peak or peaks. For example, a harmonic regression of sine wave would be mainly a single peak because it only consists of a wave at single period. From the time series of selfishness, we found oscillating dynamics at intermediate shape parameters ($a = b = 0.7$; *Appendix 1—figure 6c*), but not at low and high shape parameters (*Appendix 1—figure 6b& d*; $a = b = 0.4\,or\,1.0$). Specifically, a harmonic regression of selfishness under low and high shape parameter values gave almost horizontal lines, whereas intermediate values showed several peaks at periods around 500–1500 generations (insets of *Appendix 1— figure 6*). These peaks suggest that there is a periodic component to the evolutionary dynamics, with stochastic cycle that repeats about every 500–1500 generations, and the dynamic is not random fluctuations.

The oscillations could occur because the benefits of manipulation and selfishness become similar when shape parameters are at intermediate values, which subsequently can be influenced by the

interactions between traits. The decreased difference in benefits is supported by a broader analysis, showing the discrepancy between game-theoretical and simulated results only occurs when the former predicts selfishness is lower than the level of manipulative cheating (*Appendix 1—figure 6a*; $0.6 < a, b < 0.8$). When shape parameters are at these intermediate values, the level of selfishness in simulations does not go below the level of manipulation, as predicted by game-theoretical analysis (*Figure 5a*, dots& curves). Moreover, the simulated trait values are a lot more variable as the error bars become visible (*Figure 5a*; black vertical lines over dots).

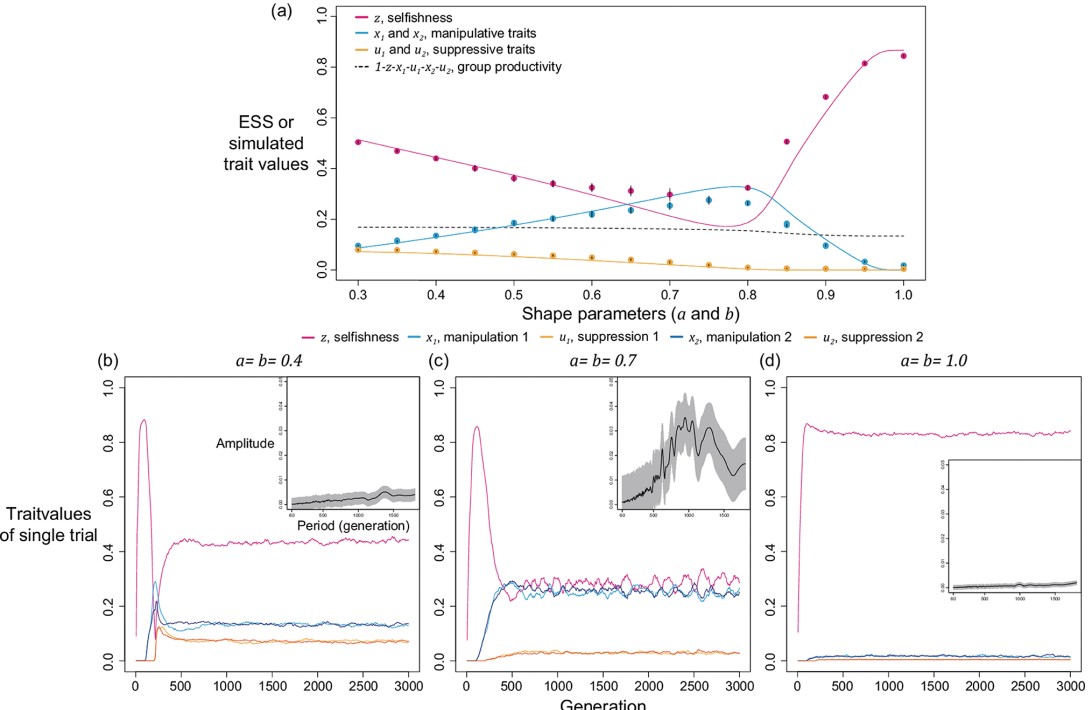

**Appendix 1—figure 6.** Oscillation dynamics of selfishness and manipulation occur in some ranges of shape parameters in the scenario with two mechanisms when relatedness is low ($R = 1/8$). (**a**) ESS predictions and simulation results from the model, where disagreement between two approaches is found when shape parameters are between 0.6 and 0.8. (**b–d**) Time series of simulations under various relatedness settings. Each panel shows the time series of traits, and the analysis from harmonic regression is shown in the inset (same as *Figure 4*). Harmonic regression found the greatest amplitude in the period of 1385 generations and p-value of $1.5333 \times 10^{-2}$ when shape parameters are 0.4; period of 815 generations and p-value of $1.8035 \times 10^{-5}$ when shape parameters are 0.7; period of 1760 and p-value of $2.5202 \times 10^{-3}$ when shape parameters are 1.0.

We can further examine the oscillations by checking the changes in maximal amplitude along with shape parameters or evolved trait values (*Appendix 1—figure 7*). In agreement with our previous results (*Appendix 1—figure 6b–d*), the maximal amplitude shows a peak when shape parameters are at intermediate values and amplitude is low when the shape is either high or low ($> 0.8$ or $< 0.6$). We are interested in the difference in evolved trait values because oscillation seems to take place when the evolved level of selfishness is close to the level at which manipulation is expressed (*Appendix 1—figure 6a*). After we visualised the relationship between trait difference and the maximal amplitude of oscillation, we found a consistent negative correlation between the difference and amplitude. This finding suggests oscillation might have resulted from interactions between traits when their evolved levels are similar.

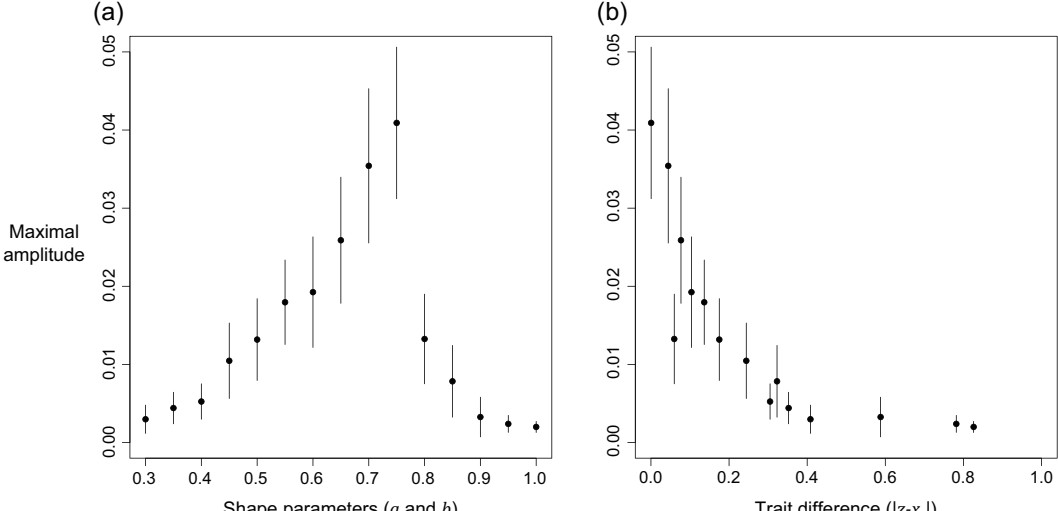

**Appendix 1—figure 7.** The relation of maximal amplitude from harmonic regression in response to (**a**) shape parameters, and (**b**) distance in evolved levels of selfishness and manipulation. Relatedness is set to 1/8 and the error bars are the 95% confidence intervals. Details of the analysis are described in Appendix 1—7.5.

## 4.3 Oscillating trait dynamics disappear when selfishness is not coevolving with manipulation

In order to better understand the origin of oscillation, we reduce the complexity in simulations by reducing the number of coevolving traits. In *Appendix 1—figure 6* and *Appendix 1—figure 7*, there are 5 traits coevolving: selfishness, manipulation 1, suppression 1, manipulation 2, and suppression 2. We make 1 of the 5 traits non-mutable at a time and analyse the evolving traits through harmonic regression. We found oscillation disappears when selfishness is held constant, while holding one of the two manipulative traits constant can still produce oscillating dynamics (*Appendix 1—figure 8b& c*). Moreover, the results are consistent regardless of whether the traits are fixed at the average of coevolving dynamics or the prediction from game-theoretical analysis (only the latter is plotted in *Appendix 1—figure 7* to avoid redundancy).

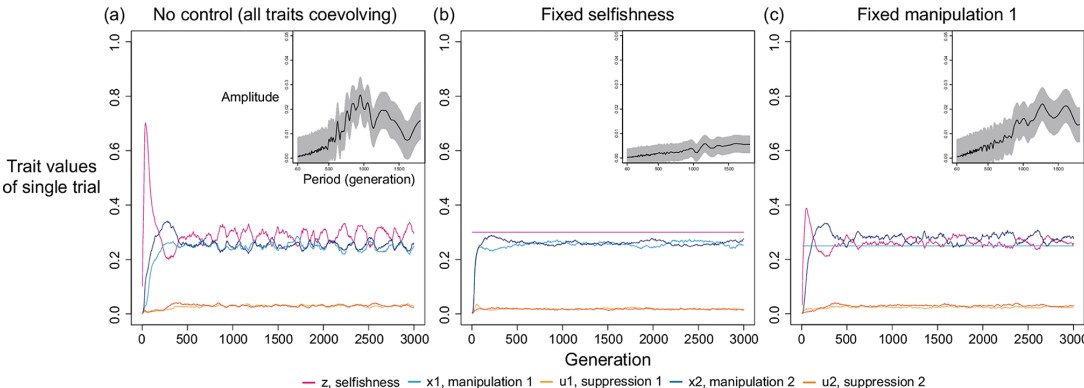

**Appendix 1—figure 8.** Oscillation disappears when selfishness is controlled and fixed at a value. We use harmonic regression to instead analyse the dynamics of manipulation 2 in this figure, because selfishness and manipulation 1 are fixed in some panels. The level of fixed selfishness and manipulation are determined by the ESS predicted in game-theoretical analysis, where relatedness is 1/8 and shape parameters are 0.7. Harmonic regression found greatest amplitude in the period of 825 generations and p-value of $2.1045 \times 10^{-4}$ when all traits coevolve; period of 1450 generations and p-value of $2.5239 \times 10^{-4}$ when selfishness is fixed; period of 1575 and p-value of $5.5099 \times 10^{-4}$ when manipulation 1 is fixed.

The above findings suggest the oscillation is likely to be an outcome from the linkage (interaction) between selfishness and manipulation. That is, some stochasticity may lead to deviations in trait values, for

instance, increased selfishness and decreased manipulation. This perturbation would generate selective forces that favours decreased level of selfishness and increased level of manipulation. However, because the optimal trait values of both traits are similar to each other, the selective force would be weak, which allows trait values to fluctuate within a range of levels. Finally, when the optimal trait values are more differed and selective forces are strong, oscillation would only take place in a small range of parameter or even disappear. Such effects of linkage (interactions) are not discovered in game-theoretical analysis because inclusive fitness effects are estimated as independent factor (i.e., no linkage between traits). Similar complication in coevolving traits have also been observed in other studies involving individual-based simulations (*Tibbetts et al., 2020*; *Liu et al., 2021*).

## 4.4 Oscillation also disappears when traits are independently inherited without linkage

To further test our linkage hypothesis for the oscillations, we examined an artificial scenario where there could be no possible linkage. Specifically, we assumed that each offspring inherits each trait independently, possibly from different individuals (multiple asexual parents, a different one for each trait). In support of our hypothesis, we found that when there can be no linkage, that oscillations do not occur (*Appendix 1—figure 9a & b*). In addition, in this scenario, where there can be no linkage, the results of the simulation where in quantitative agreement with the analytical prediction (*Appendix 1—figure 9c*). This contrasts with our simulations results in *Appendix 1—figure 6a*, where oscillations occurred, and there was a greater quantitative disagreement between our analytical and simulation results. It also suggests that some disagreement between analytical and simulation results were due to the simulation allows for correlation among inherited traits to build over time.

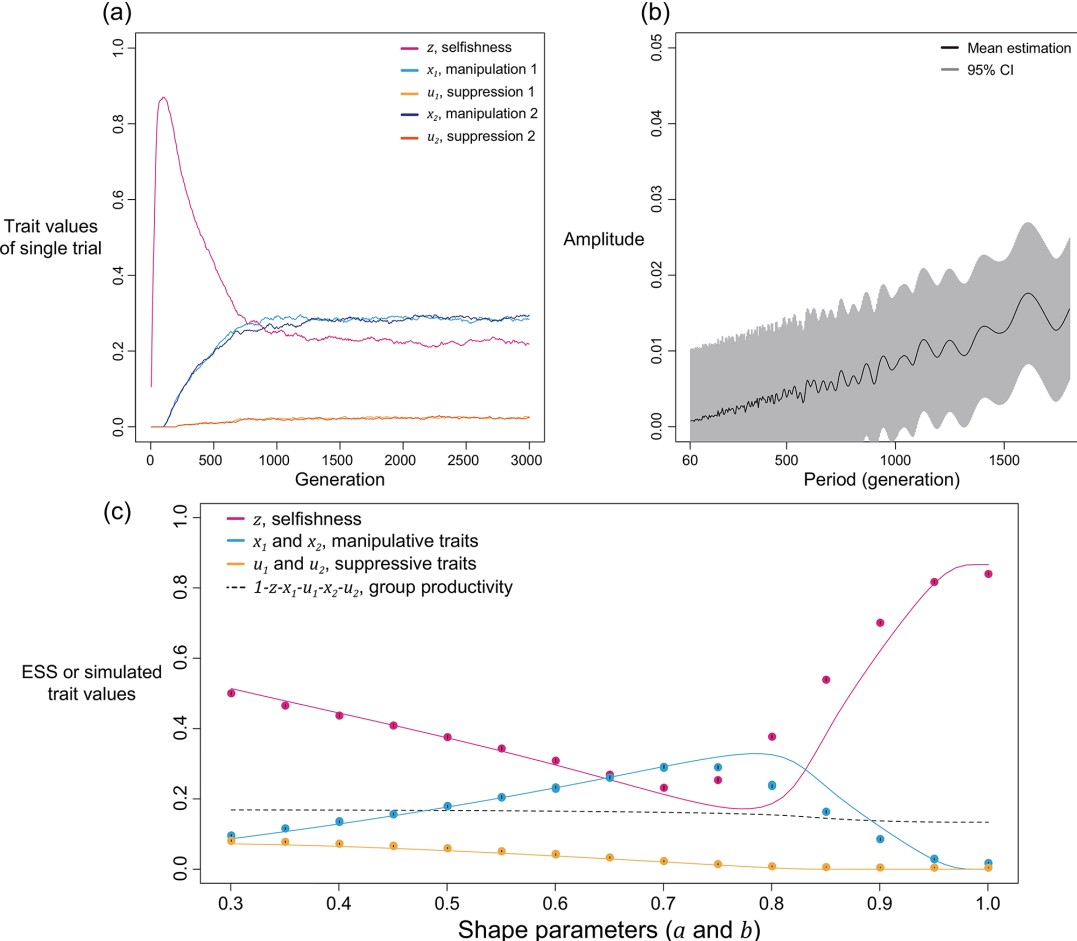

**Appendix 1—figure 9.** Oscillation does not occur when traits are inherited independently without linkage. (**a**) Time series of coevolving traits when shape parameters are intermediate, and relatedness is low ($R = 1/8$; $a = b = 0.7$). (**b**) The estimated amplitude of the time series of selfishness by harmonic regression. (**c**) Analytical predictions and simulation results from the model without linkage of traits.

## 4.5 Can oscillation happen when there are fewer or more mechanisms?

We then analyse whether oscillation can occur in cases with different number of mechanisms. The idea is if the oscillation was caused by interacting selfishness and manipulation, can oscillation still occur when there is only one mechanism to manipulate, or when there are 3 mechanisms. As shown in *Appendix 1—figure 10*, we found single-mechanism simulation cannot generate oscillation while the three-mechanism simulation still has oscillations. This makes sense because multiple mechanisms of manipulation can amplify the initially small perturbations and help oscillation to be maintained in the trait dynamics.

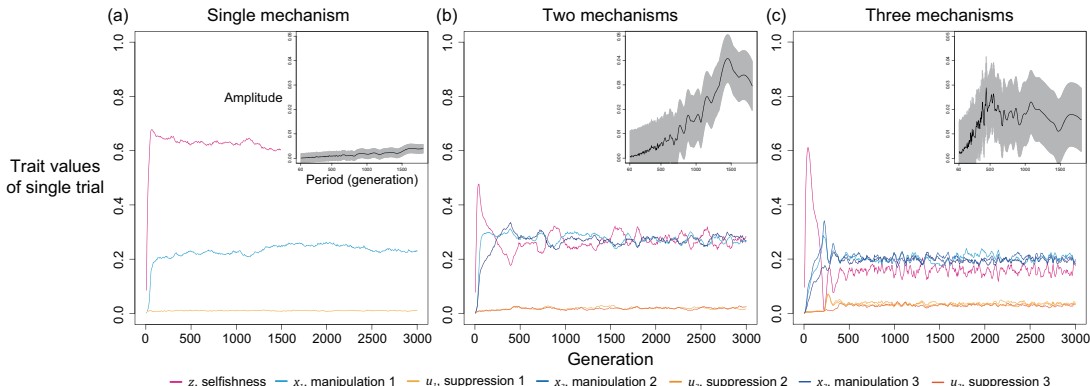

**Appendix 1—figure 10.** Oscillation happens when there is more than one mechanism to manipulate and suppress. Here, relatedness is 1/8 and shape parameters are 0.75. Harmonic regression found greatest amplitude in the period of 1800 generations and p-value of $4.7899 \times 10^{-3}$ in single-mechanism simulations; period of 1475 generations and p-value of $2.7475 \times 10^{-5}$ in two-mechanism simulations; period of 390 and p-value of $1.4544 \times 10^{-2}$ in three-mechanism simulations.

## 4.6 Does changing mutation rate lead to a different oscillation period?

In section 4.3 and 4.4 we have showed that the oscillation results from interactions and linkages between traits, consequently, we expect mutation rate should change the behaviour of oscillation. This is because mutation rate controls the speed of introducing new genotype to the population and the oscillation can be slower if mutation rate is lower. We verify this thought by running simulation with half of the original mutation rate (i.e., $5 \times 10^{-3}$, Appendix 1—7). We found supporting evidence as the oscillation of traits last longer per each cycle and have the greatest amplitude at a larger period close to 1500 generations (compare *Appendix 1—figure 11* and *Figure 4*).

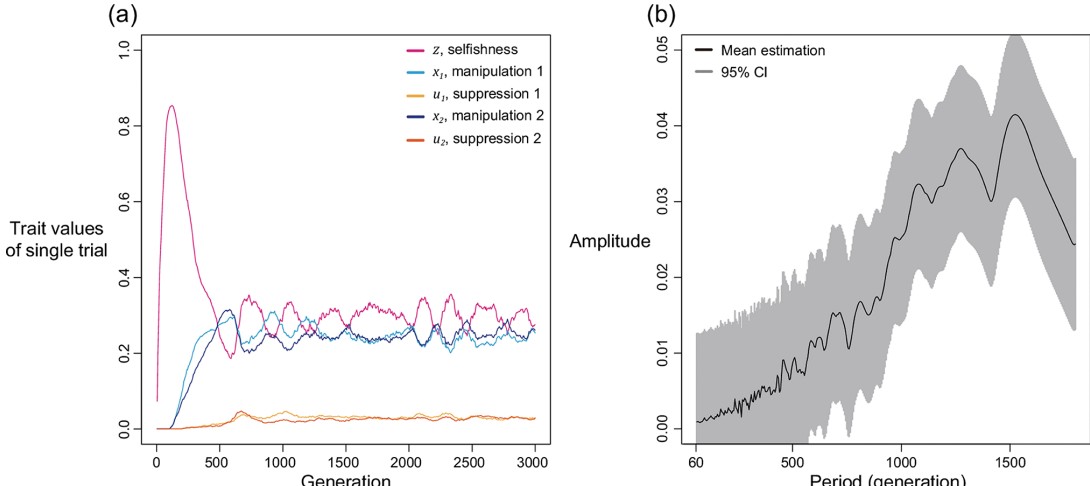

**Appendix 1—figure 11.** Selfishness and manipulative cheating traits oscillate slower when mutation rate is lower. In this figure, we set mutation rate is half the size of which in all other simulation ($R = 1/8$; $a = b = 0.7$).

## 4.7 Can we find oscillation in the personal cost model?

In section 3, we have discussed that implementing personal cost to manipulation and suppression would not qualitatively change the predictions, but can we still get oscillation in these models? We made a two-mechanism model with personal cost and found out there is no oscillation (*Appendix 1—figure 12b*). The underlying reason is the evolved level of selfishness is much higher than the level of manipulation (*Appendix 1—figure 12*). Despite the model design is different, the relation between maximal amplitude and trait difference seems to remain unchanged by comparing *Appendix 1—figure 12* and *Appendix 1—figure 7b*.

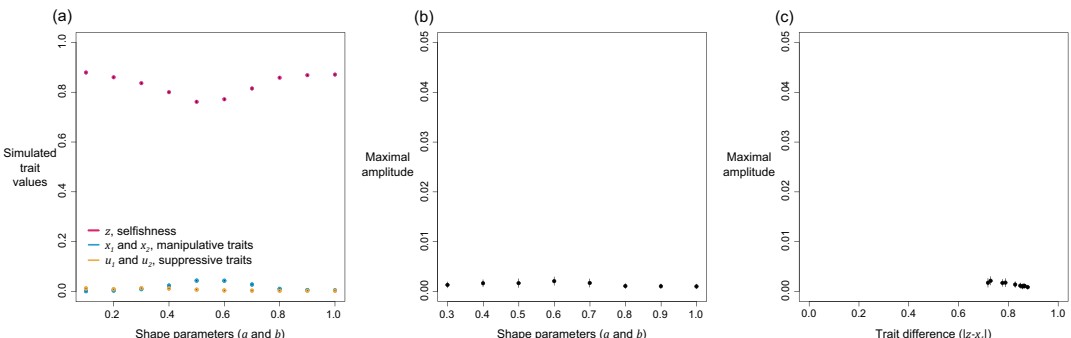

**Appendix 1—figure 12.** The properties of personal cost model with two mechanisms to manipulate and suppress. (**a**) The simulated level of selfishness, manipulative cheating, and suppression, (**b**) the relation between maximal amplitude from harmonic regression and shape parameters ($a = b$), and (**c**) the relation between maximal amplitude and trait difference in evolved selfishness and evolved manipulation.

## 4.8 Does recombination change the simulated results?

So far, we have assumed that each set of trait values are carried by single individual across generations (except Appendix 1—4.4), but it might not be the case when recombination take place, where genes may be exchanged between individual cells or segments of chromosomes swapped between two haploid individuals. We address this potential biological complexity by setting an additional recombination phase in the life cycle of individual-based simulations. During the recombination phase, individuals are selected according to relative fitness in the previous generation, paired with another, and then recombine one trait value depending upon the recombination probability (externally set parameter). Only one trait value is exchanged, and an individual recombines only once. After the possible recombination, the two individuals are put into a pool of founders. Once all individuals are in the founder pool, and a specific number of individual ($m$, Appendix 1—7) are sampled as founder to start the new population.

The effect of recombination depends on how many mechanisms of manipulation and suppression are considered in the simulation. When there is only one mechanism, we found recombination does not affect the results (*Figure 2* and *Appendix 1—figure 13a–c*). This is likely because all social interactions, including manipulation and suppression, are reset at the beginning of each generation that nothing in the past generations is passed on the future generation. When there are multiple mechanisms, recombination also plays minimal role in most cases (*Appendix 1—figure 13d–f*). However, as we have found in Appendix 1—4.4, the oscillating dynamics stop when recombination rate is above 0 (*Appendix 1—figure 13f*). Similar to our finding, recombination could matter in scenarios where associations between traits can be important and potentially build up over time, such as whether punishment can favour cooperation (*Gardner and West, 2004*; *Gardner et al., 2007*; *Lehmann et al., 2007*; *Hilbe et al., 2013*).

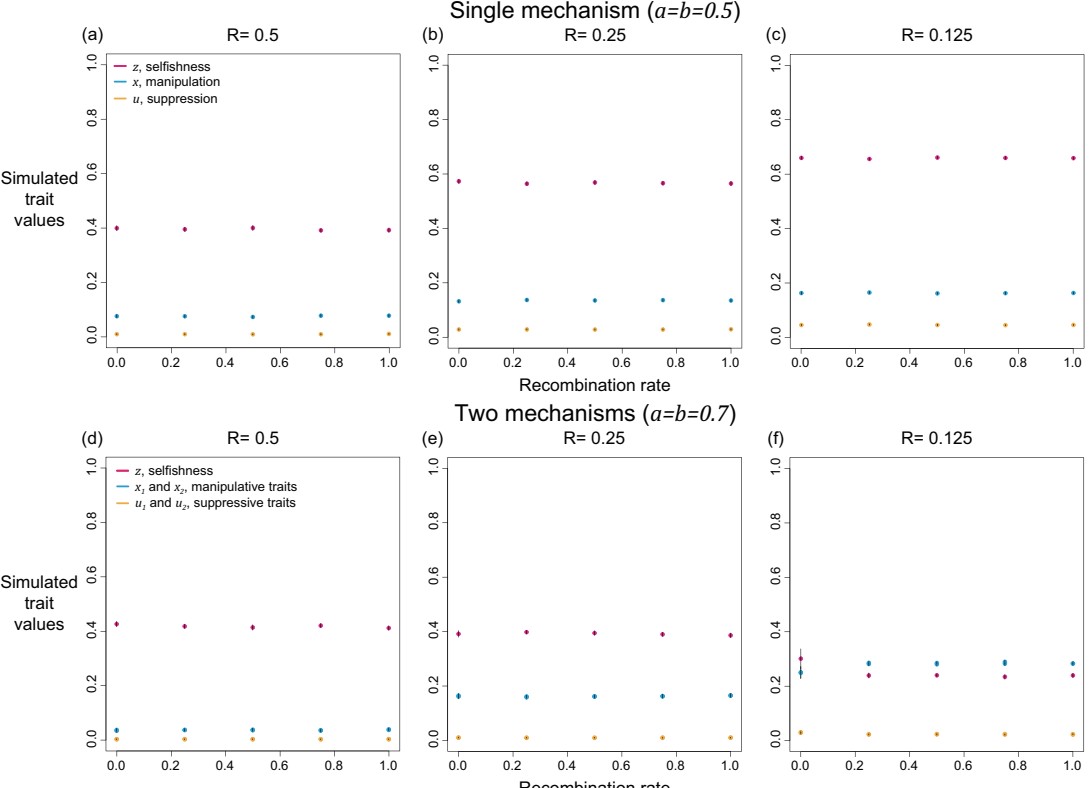

**Appendix 1—figure 13.** Effects of recombination on the simulated level of selfishness, manipulation, and suppression. Each panel has a specific relatedness setting (*R*=0.5, 0.25, or 0.125), and various recombination rates (0, 0.25, 0.5, 0.75, and 1). Two scenarios are simulated: (**a–c**) single mechanism to manipulate and suppress, (**d-f**) two mechanisms. The single mechanism results, where shape parameters are 0.5, extended the discussion in *Figure 2* (no recombination). The two mechanisms results, where shape parameters are 0.7, bridged the gap between *Appendix 1—figure 6* and *Appendix 1—figure 9* of this appendix as the former is equal to recombination rate of 0 and the latter is equal to 1.

## 5. Arms race under different biological settings

We construct two additional scenarios to understand if alternative cost design can still generate arms-race-like dynamics. In particular, we modified group productivity so that the cost of suppression is closer to a switch-on/switch-off design. We assume suppression becomes costly only when the group average is above 0.05, and the cost is the same for all suppression above 0.05. This modification allows the share of group productivity and group productivity, to disentangle with each other and relax the linkage assumed in the main model. On top of whether the cost depends continuously on the degree of suppression, the cost can be conditional and only present when there are sufficient levels of manipulation. Thus, we assume the cost of suppression is unconditional in one case, and conditional to the average level of manipulation in the other case. For simplicity, we do not randomly introduce each new mechanism but mutate them at fixed time points with equal intervals.

We found the genetic arms race of manipulation and suppression still occur in both cases, but with different dynamics of suppressive traits. When the cost is unconditional and fixed, we found all of the suppressions go to almost 1 when the shape parameter for manipulation and suppression are equal (a = b = 0.5; *Appendix 1—figure 14*). This is because the effect of suppression is relatively linear and it does not cost extra if suppression has higher values (compare *Appendix 1—figure 14c* & *Figure 5c*). When the cost is conditional to manipulation, we found a drift-like dynamics for all suppressive traits (a = 0.5, b = 0.01; *Appendix 1—figure 15*). The underlying reason is suppressions are cost-free and no longer under selection when manipulations are completely suppressed (*Appendix 1—figure 15b & c*). The cost-free suppressions are support by the dynamics of selfishness, which is not gradually decreasing like other cases (compare *Appendix 1—figure 15a* with *Figure 5a*& *Appendix 1—figure 14a*). Nevertheless, the dynamics of manipulation are very similar in all cases, thus suggesting genetic arms race is a robust result which can be found in different biological settings (*Figure 5*, *Appendix 1—figure 14*, and *Appendix 1—figure 15*).

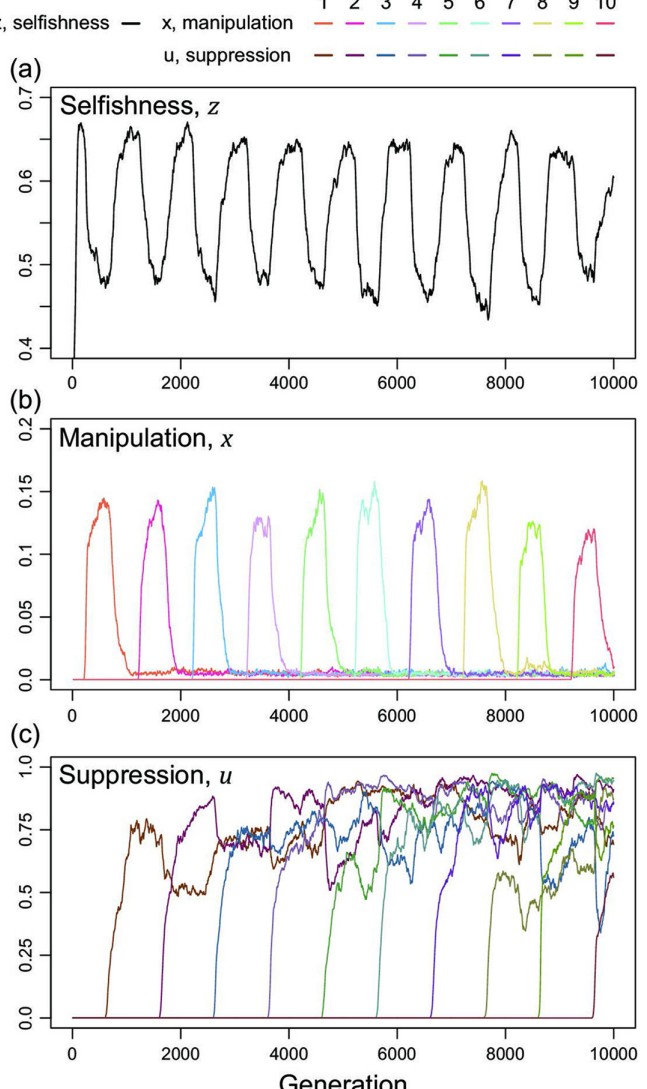

**Appendix 1—figure 14.** Arms race with an unconditional and fixed amount of cost for suppressing manipulative cheating. Suppression is costly in group productivity when it exceeds 0.05, and the cost is fixed at 0.01 regardless of the trait value of suppression. Shape parameters are equal for manipulation and suppression ($a = b = 0.5$) and relatedness is 1/3.

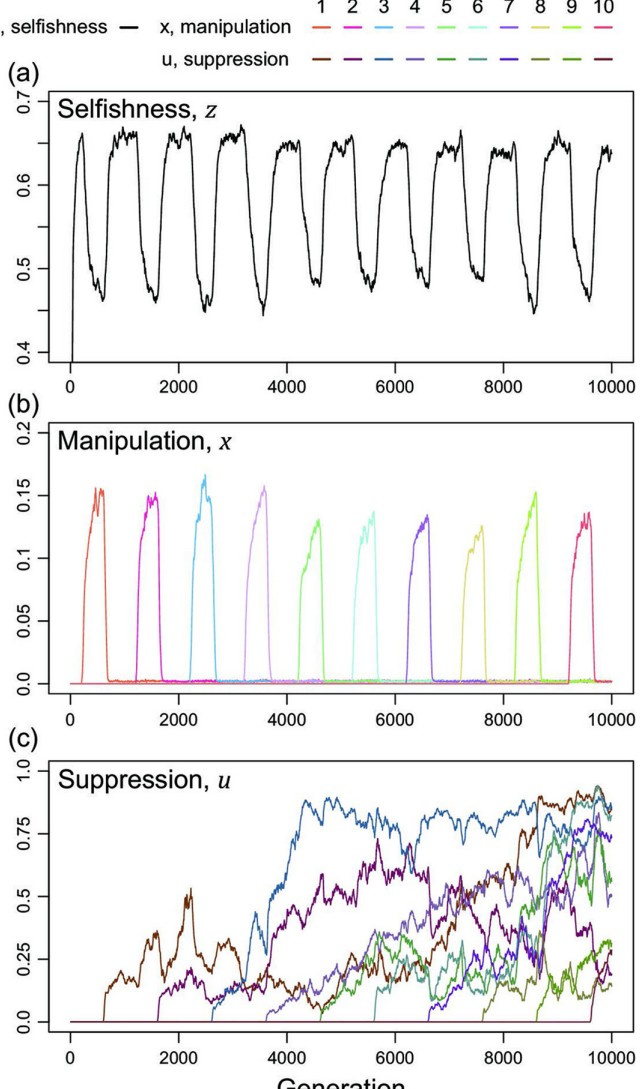

**Appendix 1—figure 15.** Arms race with a conditional and fixed cost for suppression. Suppression is costly only when both manipulative cheating and suppression are above 0.05, and the cost of suppression is fixed at 0.05 regardless of the trait value. Shape parameters are the same as *Figure 5* ($a = 0.5$ and $b = 0.01$) and relatedness is 1/3.

# 6. Additional discussion

## 6.1 Decelerating return and the nature of communication

Contrary to the linear effects assumed in the classic tragedy of the commons, empirical studies have shown that nonlinear effects might be common in communications between cells and individuals. This is because the limitation of diffusion in communicative molecules, making the benefit of increased production in those molecules diminishes (*Müller et al., 2006*; *Wiley, 2013*; *Rogers et al., 2017*); same principle can be applied to social insects, as increased time in communicating would also lead to increased possibility in encountering previously-interacted individuals (*Gill and Johnson, 2002*). As manipulation and suppression require such short-distance interaction, we have considered the nonlinear impact of traits on fitness to make the model more realistic. In particular, manipulative, and suppressive traits may have larger gains when they evolved from absence to some intermediate extent of effects, whereas the benefit of complete manipulation or suppression is diminishing. Similar saturating property has been found in other social traits, such as the benefit in increased stalk/spore ratio in slime moulds (*Bonner, 1982*; *Hudson et al., 2002*). Thus, we focus on cases where nonlinear effects have linear or diminishing returns ($0 < a \leq 1$ and $0 < b \leq 1$). For simplicity, we only add nonlinearity effect to individual reproduction component and not the group productivity component (i.e., $1 - \bar{z} - \bar{x} - \bar{u}$).

## 6.2 Methodological details of numerical analysis in game theory results

For a given number of iteration $t$, we update trait values by $\{z_{t+1}, x_{t+1}, u_{t+1}\} = \{z_t + c_{scale}\Delta W_z, x_t + c_{scale}\Delta W_x, u_t + c_{scale}\Delta W_u\}$ and find the trait values that converge to fixed values. All traits are initialised at 0.1. We set the scaling coefficient $c_{scale}$ to 2.5 x $10^{-3}$ for better precision. Finally, due to slow convergence when relatedness is low, we iterated 4 x $10^6$ times for $0.1 \geq R$ and 4 x $10^5$ times for the rest of relatedness values.

# 7. Individual-based simulations

## 7.1 Overview

We use individual-based simulations following genetic algorithm to understand the evolution of selfishness, manipulative cheating, and suppression of manipulation. In contrast to the game-theoretical analysis, individual-based simulations allow us to explicitly model mutations and trait dynamics under a continuous trait space. Following the formal analysis (section 1), we employ similar assumptions that the population size is constant, and generations do not overlap.

After initialisation, the biological processes in each generation take place in the following order: First, groups are formed from several ($l$) founders where the number of founders determines the relatedness ($R = 1/l$). Each group has the same number of founders, and it is a parameter for population structure. Second, the founders expand the social group to a size that social interactions take place, where each founder replicate the same number of times. Third, we calculate the group averages of trait values and the relative fitness of each individual. Fourth, the new-born founders are randomly drawn from the current population according to the relative fitness. Each new-born founder inherits its parent's trait values with independent chances of mutation for each of the three traits.

We use the design with founders and group growth to avoid the potential effects of small integers and small group sizes (i.e., demographic stochasticity in social groups). This simulation framework is known as haystack models (**Maynard Smith, 1964**).

## 7.2 Initialisation

We initialise the population by creating a population with uniform group size ($n$) and uniform properties of no selfish traits (i.e., $z = x = u = 0$). In order to reduce the effect of demographic stochasticity, we hold the size of the whole population and let the number of groups change because group size ($n$) is explicitly modelled in the model. Specifically, we set population size parameter ($N$) and obtain the number of groups as $M = \lceil N/n \rceil$, the least integer larger than or equal to $N/n$. Thus, the population size, $Mn$, is relatively constant while the number of groups increases as the group size decreases.

Each individual has three variables: the degree of selfishly increase own fitness share at the cost of entire group ($z$), the degree of manipulative cheating ($x$) and the degree of suppressing manipulative cheating ($u$). Within one group, there are $m = n/l$ individuals that have identical trait values because they are replicated from the same founder. Further, trait values could only be changed in *mutation* process.

## 7.3 Fitness calculation and the coevolving traits

We perform two levels of selection to choose founders for the next generation: (i) group selection according to group productivity from average trait values of all group members, and (ii) individual selection according to each group members' relative fitness (**Figure 1**). In other words, the probability that a founder comes from a group depends on the group's productivity, $1 - \bar{z} - \bar{x} - \bar{u}$, and the chance that it is an offspring of the focal individual is determined by its relative fitness in the group (or share or productivity), $\frac{z_i + (1 - u_{-i})z_{-i}x_i - (1 - u_i)z_i x_{-i}}{\bar{z}}$. Although the simulations calculate fitness in the same way as the game-theoretical analysis, the process for each event is explicitly modelled in the simulations and individuals can have any trait values, whereas the game-theoretical analysis tracks on the trait dynamics of the entire population under weak selection assumption.

## 7.4 Mutation of the coevolving traits

We assumed the traits can mutate between generations. For each trait of a new founder, the mutation rate is $p_{mut} = 0.01$. If a mutation occurs, we perturb the parental trait value by a normally distributed deviation with mean of 0 and standard deviation of 0.1, constrained at the boundaries [0,1].

## 7.5 Analysis of the simulations

In all cases, we repeated the simulation 10 times, and let social interactions take place at the population around $10^5$ individuals (i.e., $lm \lceil 10^5/lm \rceil$) to ensure that the trait values converged to their evolutionary optima. We introduced mutation to selfishness from the start of simulation, manipulative

cheating from 100 generation, and suppression from 200 generation to make sure the latter traits are not evolved in the absence of selfishness. We tracked the trait dynamics for $3\times10^3$ generations (unless specified elsewhere) and average over the last 10% to calculate the average and standard deviation of trait values. Except for the ones with oscillation dynamics, the coevolving traits usually stabilise within 500 generations (e.g., *Appendix 1—figure 6b & d*).

