## [Editor Report]

This theoretical paper examines the conditions under which manipulative cheating can evolve. Manipulative cheating is a form of coercion where individuals can benefit by actively manipulating others to help them at a cost to their own fitness. The paper seamlessly integrates a multilevel selective framework and inclusive fitness, with a rigorous analysis of the joint dynamics of selfishness, manipulation, and suppression of manipulation. The results are novel and important, as they help us to better understand the evolution of cooperation and the spectrum of social cheating, potentially opening up new directions for theoretical and empirical work.

---

## [Decision Letter]

**Decision letter after peer review:**

Thank you for submitting your article "The evolution of manipulative cheating" for consideration by *eLife*. Your article has been reviewed by three peer reviewers, and the evaluation has been overseen by Sara Mitri as the Reviewing Editor and Christian Rutz as the Senior Editor. The reviewers have opted to remain anonymous.

The reviewers have discussed their reviews with one another, and the Reviewing Editor has drafted this decision letter to help you prepare a revised submission.

Essential revisions:

The reviewers agreed that the present article introduces a potentially valuable mathematical model that can contribute to a better understanding of the role of different cheating strategies. That said, they also highlighted several shortcomings. To address these, please implement the following revisions:

1) Clearly state that the model is developed for haploid, asexual populations with no recombination. We recommend that you discuss an extension to sexual organisms in the Discussion section, including a statement on whether you expect your results to be specific to asexual populations, or generalise.

2) Clarify how you define 'manipulative cheating' mathematically and how this can be distinguished from other forms of manipulation. Use this definition to highlight the novelty of your model with respect to previously published results on manipulation. Presumably, this is -- as one of the reviewers notes -- that you are connecting manipulation and cheating, but this needs to be better spelled out.

3) Explain better the mechanism responsible for generating diversity between the different types, and how it relates to similar mechanisms documented in the literature.

4) Improve the structure and clarity of the article, emphasizing its novelty and framing it better in the context of earlier work.

5) Please also carefully address the other points that were raised in the reviewers' full reports, which are appended below.

*Reviewer #1 (Recommendations for the authors):*

1. Contrast and explain in the main text the result for a sexual population.

2. The main findings of such an analysis should be announced in the Abstract and at the end of the Introduction.

3. Instead of the simulations reported in section 4.4 of the appendix, couldn't the authors simply simulate a sexual population?

4. Two rather distinct processes of diversification are discussed in the literature: trait substitution sequences that can lead to, for example, cyclic mean trait dynamics, and frequency-dependent selection. These two phenomena are typically characterized by distinct calculations, the first case not involving any concept of frequency dependence. To help readers better understand what is being done in the manuscript, it would be helpful to place their methods in the context of theoretical frameworks for the stability of co-evolving traits, even if formal criteria for stability are not evaluated.

5. Moreover, despite sentences to the contrary, no "ESS "es are identified by the mathematical analysis, which only identifies trait values that cancel the selection gradient. For sure, it is not uncommon in the literature to use vague language, by which any type of equilibrium is called an ESS, and I am not terribly anxious that "branching" may occur here. But then again, the reader has to go through all the manuscript to make sure what is actually being calculated.

6. Provide a mathematical criterion for distinguishing manipulative cheating from other forms of manipulation. Apply it to show that the previous models were indeed not manipulative cheating models.

*Reviewer #2 (Recommendations for the authors):*

I really liked how this paper bridged MLS and inclusive fitness -- the population structure of the model was explicitly group structured and was a core driver of collective productivity (cooperation means increasing group productivity, which is precisely how the MLS models of the 80s onwards were formulated), but inclusive fitness was included in the fitness payoff equations to account for the impact of within-group social interactions.

This is a cool modeling paper with a lot to like. I thought the overall approach was quite elegant. I loved explicitly considering selfish cheating, manipulative cheating, and anti-manipulative suppression. As I said above, I liked the merging of MLS and inclusive fitness. I liked allowing for convex vs. concave payoffs to cheating and suppression. I thought the analyses were neat, and broadly made sense. Much of the detailed analysis was pushed into the appendices, which made the main paper easier to read. The theory was explained conceptually, as well as mathematically, making it easier to understand the logic of the equations.

Conceptually, I totally agree with the authors' point in the introduction that 'manipulative cheating' is not a new concept -- indeed, the term manipulation or coercion has been around for a long time, though I suspect they are right that this has so far not had much theoretical treatment. Had it just been about manipulation, I'm not sure how excited I'd be, but considering the joint evolution of manipulation with selfish cheating (the more canonical form of cheating) is really nice.

I'm not sure whether this is possible, but it would be quite nice if the authors could connect any of their model parameters, or their specific results (i.e., oscillatory dynamics), to other systems. The introduction and discussion read a bit overly general to me -- describing the fields of social evolution at a high level but not really connecting them to the specifics of this paper very well. Any opportunities to connect the results of this model to prior data or models, i.e., increased manipulation at low relatedness, anything about the actual shape parameters of cheating, data on oscillatory behavior, etc. Basically, you've shown us that you can understand manipulation with theory -- how does your work help us better understand what we know about the world around us now?

*Reviewer #3 (Recommendations for the authors):*

I have but a couple of recommendations:

- Line 105: mention Steve Franks papers right there.

- In the discussion, make a link to the theoretical/empirical work on the coevolution of offense and defense traits in ecological interactions (e.g., adaptive dynamics and such) and in sexual conflict (e.g., Gavrilets' papers).

[Editors' note: further revisions were suggested prior to acceptance, as described below.]

Thank you for resubmitting your work entitled "The evolution of manipulative cheating" for further consideration by *eLife*. Your revised article has been evaluated by Christian Rutz (Senior Editor) and Sara Mitri (Reviewing Editor), in consultation with two reviewers.

The manuscript has been improved but there are some remaining issues that need to be addressed. Specifically, please address the points raised by Reviewer #1 below, including the updating of code that is made publicly available via Github.

*Reviewer #1 (Recommendations for the authors):*

It is difficult to understand some of the revisions. In the first version, it was stated that the population is assumed to be clonal and that in the only case where this assumption is relaxed, cyclic dynamics occur (section 4.4 of the appendix). This last result is still presented in the manuscript (Figure 9 in the Appendix). Yet, new simulations are presented (Figure 13 in the Appendix) to show that recombination has no effect on the level of selfishness, manipulation, and suppression. How are these two results related? I understand that one can have different levels of oscillation despite no effect on the average trait value, but it is not clear if this is what is happening in this case.

The revised simulation code, including the new recombination procedure, is not available for inspection (the github sources have not been updated).

The authors added the statement, "There is no recombination, but our model is equivalent to one that assumes a sexual diploid with additive interactions between alleles at a given locus, with alleles at different loci segregating independently (Taylor 1996; Day and Taylor 1998)." If this statement were actually true, then oscillations should not depend on the amount of recombination. The references given in the quoted sentence cannot be used to support such claims. What do they really say about the presence or absence of oscillations for different levels of recombination? Please write a clear argument rather than a reference whose relevance is unclear.

l. 292 – "Our simulations, which involved mutations, led to relatively static predictions, in agreement with our analytical results, supporting the results as ESSs (dots versus lines in Figure 2)" and l.344 "…we found that levels of selfishness and manipulation do not always tend to a single equilibrium. Instead, they can oscillate in a periodic style […]". I cannot reconcile these two statements. Please clarify.

*Reviewer #2 (Recommendations for the authors):*

I am satisfied with the revised manuscript.

---

## [Author Response]

Essential revisions:The reviewers agreed that the present article introduces a potentially valuable mathematical model that can contribute to a better understanding of the role of different cheating strategies. That said, they also highlighted several shortcomings. To address these, please implement the following revisions:1) Clearly state that the model is developed for haploid, asexual populations with no recombination. We recommend that you discuss an extension to sexual organisms in the Discussion section, including a statement on whether you expect your results to be specific to asexual populations, or generalise.

We have followed this suggestion: (1) We added a description of population structure on line 132-137; (2) we have caried out new analyses, examining the consequences of recombination, and found that it did not alter the predictions of our model in Appendix 1—4.8; (3) we have added a brief discussion of recombination and sexual populations on line 469-475.

2) Clarify how you define 'manipulative cheating' mathematically and how this can be distinguished from other forms of manipulation. Use this definition to highlight the novelty of your model with respect to previously published results on manipulation. Presumably, this is -- as one of the reviewers notes -- that you are connecting manipulation and cheating, but this needs to be better spelled out.

We have added a definition and description, and also compared with related models (Line 187-193 and Appendix 1—1.3).

3) Explain better the mechanism responsible for generating diversity between the different types, and how it relates to similar mechanisms documented in the literature.

We have added relevant discussion on line 509-527.

4) Improve the structure and clarity of the article, emphasizing its novelty and framing it better in the context of earlier work.

We have attempted to clarify the structure and clarity, while also trying to balance when the suggestions differ between reviewers.

5) Please also carefully address the other points that were raised in the reviewers' full reports, which are appended below.

We have addressed the points as detailed below.

Reviewer #1 (Recommendations for the authors):1. Contrast and explain in the main text the result for a sexual population.

We have carried out additional analyses with recombination, which are discussed on lines 132-137, 469-475 and in Appendix 1—4.8. By assuming additive effects of traits and each traits having the same dispersal property, our haploid asexual model is also equivalent to a diploid sexual model (Taylor 1996; Day and Taylor 1998).

2. The main findings of such an analysis should be announced in the Abstract and at the end of the Introduction.

We have added a description of this analysis to the introduction, on lines 90-91.

3. Instead of the simulations reported in section 4.4 of the appendix, couldn't the authors simply simulate a sexual population?

As discussed in point 1, we have added extra analyses, examining the consequences of recombination (Appendix 1—4.8), some discussion on sexual and asexual populations, and linkages to previous theory papers from Peter Taylor on these two types of populations on line 472-474.

4. Two rather distinct processes of diversification are discussed in the literature: trait substitution sequences that can lead to, for example, cyclic mean trait dynamics, and frequency-dependent selection. These two phenomena are typically characterized by distinct calculations, the first case not involving any concept of frequency dependence. To help readers better understand what is being done in the manuscript, it would be helpful to place their methods in the context of theoretical frameworks for the stability of co-evolving traits, even if formal criteria for stability are not evaluated.

As suggested, we have made the links to these theoretical literatures about trait substitution sequences at lines 415-417 with the results in Figure 5, and the discussion about frequency-dependent selection on lines 364-371 to the results in

Figure 4.

5. Moreover, despite sentences to the contrary, no "ESS "es are identified by the mathematical analysis, which only identifies trait values that cancel the selection gradient. For sure, it is not uncommon in the literature to use vague language, by which any type of equilibrium is called an ESS, and I am not terribly anxious that "branching" may occur here. But then again, the reader has to go through all the manuscript to make sure what is actually being calculated.

Thanks, we have added statements to clarify on line 286-294.

6. Provide a mathematical criterion for distinguishing manipulative cheating from other forms of manipulation. Apply it to show that the previous models were indeed not manipulative cheating models.

We have added a detailed mathematical comparison in Appendix 1—1.3, and further discussion on line 187-193.

Reviewer #2 (Recommendations for the authors):I really liked how this paper bridged MLS and inclusive fitness -- the population structure of the model was explicitly group structured and was a core driver of collective productivity (cooperation means increasing group productivity, which is precisely how the MLS models of the 80s onwards were formulated), but inclusive fitness was included in the fitness payoff equations to account for the impact of within-group social interactions.This is a cool modeling paper with a lot to like. I thought the overall approach was quite elegant. I loved explicitly considering selfish cheating, manipulative cheating, and anti-manipulative suppression. As I said above, I liked the merging of MLS and inclusive fitness. I liked allowing for convex vs. concave payoffs to cheating and suppression. I thought the analyses were neat, and broadly made sense. Much of the detailed analysis was pushed into the appendices, which made the main paper easier to read. The theory was explained conceptually, as well as mathematically, making it easier to understand the logic of the equations.Conceptually, I totally agree with the authors' point in the introduction that 'manipulative cheating' is not a new concept -- indeed, the term manipulation or coercion has been around for a long time, though I suspect they are right that this has so far not had much theoretical treatment. Had it just been about manipulation, I'm not sure how excited I'd be, but considering the joint evolution of manipulation with selfish cheating (the more canonical form of cheating) is really nice.

Many thanks for all your positive feedbacks, really appreciated!

I'm not sure whether this is possible, but it would be quite nice if the authors could connect any of their model parameters, or their specific results (i.e., oscillatory dynamics), to other systems. The introduction and discussion read a bit overly general to me -- describing the fields of social evolution at a high level but not really connecting them to the specifics of this paper very well. Any opportunities to connect the results of this model to prior data or models, i.e., increased manipulation at low relatedness, anything about the actual shape parameters of cheating, data on oscillatory behavior, etc. Basically, you've shown us that you can understand manipulation with theory -- how does your work help us better understand what we know about the world around us now?

Our model was designed to capture a general principle that could apply across species, rather than quantitative predictions for a single species. We believe it helps us understand at two levels. First, it shows how more complex forms of cheatings can be favoured. We have discussed this level, with examples, on lines 477-492. Second, it can explain the more specific observation of genetic variation in cheating / suppression in ants and bacteria. We have added a specific discussion of this on lines 427-430.

Reviewer #3 (Recommendations for the authors):I have but a couple of recommendations:- Line 105: mention Steve Franks papers right there.- In the discussion, make a link to the theoretical/empirical work on the coevolution of offense and defense traits in ecological interactions (e.g., adaptive dynamics and such) and in sexual conflict (e.g., Gavrilets' papers).

Many thanks for these useful suggestions – we have made changes accordingly (Line 107-108 and 517-521).

References

1. Day, T. and Taylor, P.D. (1998). Unifying Genetic and Game Theoretic Models of Kin Selection for Continuous Traits. *Journal of Theoretical Biology*, 194, 391-407.

2. Taylor, P.D. (1996). Inclusive fitness arguments in genetic models of behaviour. *Journal of Mathematical Biology*, 34, 654-674.

[Editors' note: further revisions were suggested prior to acceptance, as described below.]

The manuscript has been improved but there are some remaining issues that need to be addressed. Specifically, please address the points raised by Reviewer #1 below, including the updating of code that is made publicly available via Github.Reviewer #1 (Recommendations for the authors):It is difficult to understand some of the revisions. In the first version, it was stated that the population is assumed to be clonal and that in the only case where this assumption is relaxed, cyclic dynamics occur (section 4.4 of the appendix). This last result is still presented in the manuscript (Figure 9 in the Appendix). Yet, new simulations are presented (Figure 13 in the Appendix) to show that recombination has no effect on the level of selfishness, manipulation, and suppression. How are these two results related? I understand that one can have different levels of oscillation despite no effect on the average trait value, but it is not clear if this is what is happening in this case.

Thanks for pointing out the discrepancy in the effects of recombination in the Appendix. The numbers of mechanisms for manipulation and suppression were different between Figure 9 and Figure 13 in the Appendix, where two mechanisms were simulated in Figure 9 and one mechanism was simulated in the old version of Figure 13. In other words, our discussion on the old Figure 13 that recombination has no effect was about cases where no oscillation is never possible because there is only one mechanism to manipulate or suppress manipulation.

We have made a new version of Figure 13 in the Appendix to clarify, with three additional panels to show recombination changes some simulation results when multiple mechanisms are considered (Figure 13d-f in the Appendix). We have also changed the texts accordingly, in both results and the Appendix (Line 399-413, and 1364-1380). These analyses showed that a small degree of recombination is sufficient to remove oscillations in the two-mechanisms case, while recombination has no effect in single-mechanism case, where oscillation is never possible to happen. In addition, when recombination rate is at 1 in Figure 13f, the result repeats Figure 9 in the Appendix.

The revised simulation code, including the new recombination procedure, is not available for inspection (the github sources have not been updated).

We have uploaded the latest code to GitHub, sorry that it wasn’t done in the last round of revision. The code for simulating recombination is available at:

https://github.com/mingpapilio/Codes_Manipulative_Cheat/tree/main/code/sim ulative_recombination; and the data for making Figure 13 has also been uploaded.

The authors added the statement, "There is no recombination, but our model is equivalent to one that assumes a sexual diploid with additive interactions between alleles at a given locus, with alleles at different loci segregating independently (Taylor 1996; Day and Taylor 1998)." If this statement were actually true, then oscillations should not depend on the amount of recombination. The references given in the quoted sentence cannot be used to support such claims. What do they really say about the presence or absence of oscillations for different levels of recombination? Please write a clear argument rather than a reference whose relevance is unclear.

We have made a new paragraph in model design to address the assumptions on recombination in different analysis (Line 206-218). We have also added some discussion with more specific references on the effects of recombination in results (Line 399-413). In our latest analysis, we found oscillation is absence in all non-zero recombination rates and only presence when there is no recombination (Figure 13f in the Appendix). This finding coincides with previous literature that recombination prevents evolutionary branching into distinct genotypes, because it removes the linkage between strategies (linkage disequilibrium) (Abrams *et al.* 1993; Dieckmann and Doebeli 1999).

l. 292 – "Our simulations, which involved mutations, led to relatively static predictions, in agreement with our analytical results, supporting the results as ESSs (dots versus lines in Figure 2)" and l.344 "…we found that levels of selfishness and manipulation do not always tend to a single equilibrium. Instead, they can oscillate in a periodic style […]". I cannot reconcile these two statements. Please clarify.

Thanks for pointing out the issue, we have modified the former text by specifying the agreement happens when only one mechanism of manipulation and suppression is involved (Line 307-308). We have also added a sentence to clarify the ESS concept we use throughout the text is convergently stable, candidate ESS, or comparative statics, which was first framed by Steven Frank (Line 251-254).

References

1. Abrams, P.A., Matsuda, H. & Harada, Y. (1993). Evolutionarily unstable fitness maxima and stable fitness minima of continuous traits. *Evolutionary Ecology*, 7, 465-487.

2. Dieckmann, U. & Doebeli, M. (1999). On the origin of species by sympatric speciation. *Nature*, 400, 354-357.